# Localized Operator Learning with Adaptive Partition-of-Unity Mixture-of-Expert Networks

## Abstract

Operator learning methods such as DeepONets and FNOs often struggle on PDE families with sharp interfaces, heterogeneous coefficients, and localized multiscale structure. We introduce a partition-of-unity (POU) mixture-of-experts framework for localized operator learning, in which geometry-aware gating networks produce smooth spatial partitions and route computation to specialized local experts. Our main contribution is HiRefPOU, a hierarchical residual POU architecture for DeepONets that enables coarse-to-fine refinement while preserving global continuity. We also show that the same POU principle can be incorporated into Fourier Neural Operators to introduce spatial adaptivity without modifying the underlying spectral layers. On heterogeneous Darcy and reaction–diffusion benchmarks, HiRefPOU achieves substantially lower error than global DeepONet and static POU-MoE baselines, while the broader operator-learning experiments show that the benefits of localization depend on the PDE structure and the chosen neural-operator backbone. The learned partitions are interpretable and align with interfaces and regions of rapid solution variation. These results show that explicit geometric localization can improve both accuracy and interpretability in neural operator learning.

## 1 Introduction

The emergence of machine learning (ML)-based partial differential equation (PDE) solvers has enabled the efficient and data-driven modeling of complex physical systems. Among these are neural operator frameworks, such as Deep Operator Networks (DeepONets) (Lu et al., 2021) and Fourier Neural Operators (FNOs) (Li et al., 2020), which learn mappings between function spaces and capture the input-output behavior of entire PDE families. These models have achieved notable success across a range of domains, including fluid dynamics, porous media flow, and materials science, where scalable operator-learning approaches are increasingly central to practical surrogate modeling in scientific computing.

Neural operators, while designed to learn mappings between families of PDEs, can encounter limitations when approximating solutions with localized, multi-scale, or heterogeneous structures. In practice, many neural operator architectures rely on globally supported components, such as dense neural networks defined across the full problem domain or spectral expansions, which tend to under-represent high-frequency or spatially localized features. While such global formulations preserve smoothness and continuity, they limit the network's ability to adaptively allocate resolution in regions of higher solution complexity, often resulting in an insufficient representation of fine-scale structure, leading to reduced accuracy in these areas.

A natural remedy is to introduce locality through domain decomposition or mixture-of-experts formulations. In the physics-informed setting, methods such as APINNs (Hu, 2023), XPINNs (Jagtap & Karniadakis, 2020), and POU-PINNs (Rodriguez et al., 2024) decompose the domain into subregions, while recent ensemble DeepONet models (Sharma & Shankar, 2025) introduce partition-of-unity (POU) mixtures on the operator-learning side. These approaches demonstrate the value of localization, but they are typically fixed, single-level, or only weakly structured geometrically, and therefore do not provide adaptive hierarchical refinement for neural operator learning.

On the operator-learning side, ensemble and mixture-based formulations have emerged for introducing locality while preserving a global input–output structure. In particular, recent ensemble DeepONet architectures (Sharma & Shankar, 2025) decompose the trunk network (i.e., the evaluation-coordinate encoder) into a mixture of expert components satisfying a partition-of-unity (POU) constraint, and demonstrate that combining multiple operator experts improves expressivity relative to both single global models and comparably parameterized baselines. These results provide strong motivation for incorporating geometrically structured ensembles into neural operator models. However, these approaches remain limited to flat (single-level) decompositions and do not support adaptive or hierarchical refinement across multiple spatial scales.

POU constructions provide a natural mechanism for blending multiple localized models through smooth, overlapping weighting functions by introducing learnable partition functions that modulate the contributions of local subnetworks (Lee et al., 2021). Existing neural POU approaches have primarily been explored in the context of single-function approximation and have seen more limited development in operator-learning settings. Moreover, these methods typically rely on flat partitioning schemes and do not jointly incorporate adaptive partitioning, geometric awareness, and expressive neural parameterizations within a hierarchical framework.

A complementary line of work introduces hierarchical refinement through residual network architectures, which improve accuracy in PDE problems with sharp gradients by incrementally correcting coarse global approximations. Hybrid Residual PINNs (HyResPINNs) (Cooley et al., 2025) exemplify this approach by refining a global solution using localized residual components. While effective, such residual architectures rely on predefined combinations of neural and RBF-based components, leading to fixed, radially structured regions of influence and limiting the ability to form flexible, data-adaptive partitions or hierarchical mixture-of-experts decompositions. Taken together, these observations motivate the extension of hierarchical decomposition principles from solution learning to operator learning.

We address this limitation with a hierarchical POU framework for operator learning that combines smooth learnable partitions, localized experts, and residual coarse-to-fine refinement. The resulting architecture preserves global continuity while allowing targeted local specialization where the operator is hardest to approximate.

Although the proposed framework is also relevant to physics-informed settings, the main contributions of this paper are in localized neural operator learning. Our architectural developments, mathematical formulation, and numerical studies focus on DeepONet- and FNO-type models, while PINN-related discussions are included primarily for motivation and conceptual contrast.

Specifically, this paper makes the following contributions:

- We propose HiRefPOU, a hierarchical partition-of-unity framework for localized neural operator learning.

- We introduce a hierarchical POU architecture and study one-, two-, and three-level instantiations, with the two-level formulation serving as the primary architectural case and the three-level variant used to probe deeper refinement.

- We embed the proposed POU mechanism in DeepONet and FNO architectures and evaluate it on a range of operator-learning benchmarks, with a targeted PINN case study included in the appendix.

- Across these experiments, we show that hierarchical POU models improve accuracy and geometric interpretability relative to global and fixed-POU baselines, with clear accuracy–cost tradeoffs.

The remainder of this paper is organized as follows. Section 2 reviews related work in physics-informed and operator-learning architectures. Section 3 introduces the relevant background related to PINNs, operator learning, and hierarchical POU models and their mathematical formulation. Section 4 presents the hierarchical POU architecture and geometry-aware gating formulation. Section 5 describes its instantiation in DeepONet and FNO architectures and analyzes computational complexity. Section 6 outlines the numerical experiments,

baseline methods and relevant implementation details, while Section 7 discusses results and Section 8 concludes.

## 2 Related Works

This section reviews prior work most relevant to the proposed hierarchical partition-of-unity (POU) framework: physics-informed neural networks, neural operators, domain decomposition and POU methods, mixture-of-experts architectures, and hierarchical neural representations. Our emphasis is on localized operator learning, which is the primary methodological and empirical focus of this paper.

### 2.1 Physics-Informed Neural Networks (PINNs)

In the context of PDE solving, PINNs represent the most direct attempt to approximate physical laws through global neural surrogates (Raissi et al., 2017a;b) and approximate the solutions of PDEs by embedding physical laws directly into the neural network's loss function. They offer a flexible, mesh-free framework in which expressivity arises from the architecture, activation functions, and optimization process (Hornik et al., 1989; Cybenko, 1989). While universal approximation theorems establish that sufficiently large networks can represent any continuous function, realizing this theoretical expressivity in practice remains challenging (DeVore et al., 2020; Adcock & Dexter, 2021; Marwah et al., 2021). In particular, standard PINNs employ globally coupled representations, where parameters are updated according to a loss aggregated over the entire domain. This implicit, global adaptivity contrasts sharply with the localized refinement mechanisms used in classical numerical methods and limits the network's ability to capture sharp gradients, discontinuities, or multiscale behaviors (Wang et al., 2022).

### 2.2 Neural Operator Learning

Neural operators learn mappings between function spaces and provide data-driven surrogates for PDE families. Two prominent examples are DeepONets (Lu et al., 2021), which combine branch and trunk networks to approximate nonlinear operators, and Fourier Neural Operators (Li et al., 2020), which learn operators through spectral convolutions. Despite their success, standard neural operators remain largely global representations. DeepONets rely on global trunk bases, while FNOs rely on globally supported Fourier modes; both can struggle with localized, heterogeneous, or multiscale structure. Recent extensions introduce multiresolution kernels, adaptive discretizations, or localized spectral filters, but these approaches often remain computationally expensive or lack clear geometric interpretability. This motivates architectures that combine global operator structure with adaptive spatial localization.

### 2.3 Domain Decomposition and Partition of Unity Methods

Recent extensions of PINNs mitigate optimization and scalability challenges through domain decomposition, which divides the computational domain into smaller subregions for localized approximation and parallelization. Methods such as XPINNs (Jagtap & Karniadakis, 2020) and DT-PINNs (Sharma & Shankar, 2022) still rely largely on fixed or manually designed partitions, while soft decomposition approaches use adaptive weighting schemes to reduce hand-crafted partitioning (Barbara et al., 2024; Hu, 2023). These results highlight the value of learnable localization, but most existing approaches remain non-hierarchical and use standard MLP-based gating.

Partition-of-unity (POU) methods provide a classical way to build globally continuous approximations from overlapping local components. In finite element and meshfree settings, the POU constraint enables local adaptivity while preserving global consistency, making it a natural foundation for machine-learning-based PDE solvers. Unlike xPINN-style methods, which train separate models on disjoint subdomains and enforce consistency through interface penalties, our POU formulation constructs a single global approximation as a weighted sum of local experts. The weights are learned jointly with the solution, remain continuous, and sum to one over the domain, so continuity is enforced implicitly while the partitions adapt to solution structure.

### 2.4 Mixture-of-Experts and Adaptive Gating Architectures

Mixture-of-experts (MoE) models provide an adaptive counterpart to domain decomposition by representing a global approximation as a weighted combination of specialized subnetworks. In scientific machine learning, this allows different experts to focus on distinct spatial regions, physical regimes, or modes of variation while remaining coupled through a shared training objective. Related ideas have appeared in both PINN and operator-learning settings, where learnable gating is used to decompose complex mappings into simpler local components (Hu, 2023; Wang et al., 2025; Sharma & Shankar, 2025). In particular, the ensemble DeepONet framework of Sharma & Shankar (2025) demonstrates that partitioning the trunk representation into localized experts can improve expressivity and accuracy relative to purely global operator networks, providing direct motivation for localized operator learning.

Despite this promise, standard MoE formulations remain limited for PDE and operator-learning problems. Most gating networks are implemented as generic neural modules with little explicit geometric structure, making the learned partitions difficult to interpret and not necessarily aligned with meaningful physical subregions (Lee et al., 2021). Moreover, gating is often performed in latent feature space rather than directly in the spatial coordinates, which can hinder spatial coherence and reduce control over how experts divide the domain. Even structured alternatives such as RBF-based gating typically impose purely radial supports, which can be too restrictive for anisotropic or irregular localized behavior. These limitations suggest that adaptive specialization alone is not sufficient: for PDE and operator-learning problems, the gating mechanism should also be geometry-aware and spatially interpretable.

The framework proposed in this work builds on this line of research by introducing a geometrically structured MoE formulation for operator learning. Our gating functions are defined through learnable centers and distance-based features, then refined through neural corrections, yielding partitions that remain continuous, adaptive, and interpretable throughout training. In this sense, our approach may be viewed as extending localized operator-learning ideas such as Sharma & Shankar (2025) from a flat expert decomposition to a geometry-aware partition-of-unity architecture that is explicitly designed for adaptive spatial localization.

### 2.5 Multi-Fidelity and Hierarchical Neural Architectures

Hierarchical and multi-fidelity architectures address a complementary limitation of global neural PDE models: many problems exhibit structure across multiple spatial or temporal scales and therefore benefit from coarse-to-fine refinement. This principle is classical in numerical analysis, where multigrid, wavelet, and adaptive finite element methods improve approximation quality by combining coarse global structure with targeted fine-scale correction (Gerhard et al., 2022; Restrepo & Leaf, 1995; Mitchell, 1991). Analogous ideas have emerged in scientific machine learning through multi-level PINNs, multi-fidelity surrogates, and hierarchical neural operators, where progressively refined subnetworks or feature maps improve accuracy and stability (Howard et al., 2023).

Among the most closely related ideas are residual hierarchical architectures, which refine a coarse approximation through additional localized correction terms. For example, HyResPINNs (Cooley et al., 2025) combine global and local components in a residual hierarchy to improve approximation of PDE solutions with sharp gradients and heterogeneous structure. This provides strong evidence that hierarchical refinement can be beneficial in scientific machine learning. However, such approaches are primarily designed for solution learning rather than operator learning, and their local regions of influence are typically induced by predefined architectural choices rather than by an adaptive partition-of-unity decomposition learned jointly from data.

More broadly, most existing hierarchical neural architectures operate over fixed scales or latent feature hierarchies rather than over learned geometric partitions of the physical domain. As a result, they provide multiscale refinement without explicitly learning where localized specialization should occur. This is the main distinction we emphasize in the present work. Our framework combines hierarchical residual refinement with geometry-aware partition-of-unity gating, yielding a model in which coarse-to-fine refinement is organized through learned spatial partitions rather than only through global or feature-space multiscale representations. Viewed this way, the proposed method bridges two strands of prior work: localized expert decompositions

for operator learning (Sharma & Shankar, 2025) and residual hierarchical refinement in scientific machine learning (Cooley et al., 2025).

**Summary.**  Existing approaches typically provide either adaptive expert specialization without strong geometric structure, or hierarchical refinement without learned spatial partitioning. In contrast, the proposed hierarchical POU framework combines geometry-aware, learnable partitions with residual coarse-to-fine refinement in a single operator-learning architecture. This yields a model that is globally continuous, spatially interpretable, and adaptively localized across multiple scales, linking localized expert decompositions in operator learning (Sharma & Shankar, 2025) with residual hierarchical refinement in scientific machine learning (Cooley et al., 2025).

## 3    Preliminaries

### 3.1    Problem Setup

We consider PDE solution approximation and, more generally, operator learning between function spaces. Let $\mathcal{D} = [0, T] \times \Omega \subset \mathbb{R}^{1+d}$, where $\Omega \subset \mathbb{R}^d$ is a bounded spatial domain with boundary $\partial\Omega$. A general time-dependent PDE takes the form

$$u_t + \mathcal{F}(u, \nabla u, \dots) = f, \tag{1}$$

subject to initial and boundary conditions $u(0, \mathbf{x}) = g(\mathbf{x})$ for $\mathbf{x} \in \Omega$ and $\mathcal{B}[u(t, \mathbf{x})] = 0$ for $t \in [0, T]$ and $\mathbf{x} \in \partial\Omega$.

Operator learning seeks to approximate a mapping $\mathcal{G} : \mathcal{U} \to \mathcal{V}$ from admissible inputs $a \in \mathcal{U}$ to solution functions $s \in \mathcal{V}$. We write the governing problem abstractly as

$$\mathcal{N}(a, s) = 0, \qquad \mathcal{B}(a, s) = 0,$$

and assume that each admissible $a$ defines a unique solution $s = s(a)$, inducing the solution operator $\mathcal{G} : a \mapsto s$.

Given paired data $\{a^{(i)}, s^{(i)}\}_{i=1}^N$, the goal is to learn $\mathcal{G}_\theta \approx \mathcal{G}$ so that unseen inputs can be mapped to their corresponding solutions without solving the PDE from scratch. DeepONets and FNOs implement this idea through different parameterizations of the input-function representation and output evaluation map.

### 3.2    Physics-Informed Neural Networks (PINNs)

The general formulation of PINNs (Raissi, 2018) aims to approximate the unknown solution $u(t, \mathbf{x})$ using a parameterized model $u_\theta(t, \mathbf{x})$, such that $u_\theta(t, \mathbf{x}) \approx u(t, \mathbf{x})$ and $\theta$ denotes the set of all trainable parameters of the network. The parameters $\theta$ are optimized to minimize the composite loss function:

$$L(\theta) = \lambda_{ic} L_{ic} + \lambda_{bc} L_{bc} + \lambda_r L_r, \tag{2}$$

where $L_{ic}$, $L_{bc}$, and $L_r$ represent the loss components associated with the initial conditions, boundary conditions, and the residual of the PDE, respectively; and $\lambda_{ic}$, $\lambda_{bc}$, and $\lambda_r$ are corresponding weighting coefficients that balance the contribution of each loss term. These terms can be set as static weights or adapted during training using various techniques such as NTK weighting (Wang et al., 2021b) or gradient annealing (Wang et al., 2021a), among others.

Each loss term in Equation (2) is computed as the mean squared error of the initial condition, boundary, and PDE residuals. Specifically, each loss term is defined as:

$$L_{ic} = \frac{1}{N_{ic}} \sum_{i=1}^{N_{ic}} \left| u_\theta(0, \mathbf{x}_{ic}^i) - g(\mathbf{x}_{ic}^i) \right|^2, \tag{3}$$

$$L_{bc} = \frac{1}{N_{bc}} \sum_{i=1}^{N_{bc}} \left| \mathcal{B}[u_\theta](t_{bc}^i, \mathbf{x}_{bc}^i) \right|^2, \tag{4}$$

$$L_r = \frac{1}{N_r} \sum_{i=1}^{N_r} \left| \frac{\partial u_\theta}{\partial t}(t_r^i, \mathbf{x}_r^i) + \mathcal{F}[u_\theta](t_r^i, \mathbf{x}_r^i) - f(t_r^i, \mathbf{x}_r^i) \right|^2. \tag{5}$$

The training data points $\{\mathbf{x}_{ic}^i\}_{i=1}^{N_{ic}}$, $\{t_{bc}^i, \mathbf{x}_{bc}^i\}_{i=1}^{N_{bc}}$ and $\{t_r^i, \mathbf{x}_r^i\}_{i=1}^{N_r}$ can be drawn from a fixed set or resampled randomly at each iteration of a gradient descent algorithm.

### 3.3 Deep Operator Networks (DeepONets)

Deep Operator Networks (DeepONets) are designed to approximate nonlinear operators $\mathcal{G} : \mathcal{U} \to \mathcal{V}$ by decomposing the mapping into two subnetworks: the branch network and the trunk network. The branch network $B_\theta$ takes as input a finite-dimensional representation of the function $a \in \mathcal{U}$, for example

$$\mathbf{a} = (a(x_1), \dots, a(x_m))$$

evaluated at fixed sensor locations $\{x_i\}_{i=1}^m$, and produces a latent coefficient vector

$$\mathbf{b}(a) = \big(b_1(\mathbf{a}), b_2(\mathbf{a}), \dots, b_q(\mathbf{a})\big)^\top \in \mathbb{R}^q.$$

The trunk network $T_\phi$ takes as input the evaluation coordinate $y \in \mathbb{R}^p$ and outputs a latent basis vector,

$$\mathbf{t}(y) = \big(t_1(y), t_2(y), \dots, t_q(y)\big)^\top \in \mathbb{R}^q.$$

The output of the DeepONet for a given $a$ and $y$ is obtained by merging the branch and trunk outputs through an inner product:

$$G_\theta(a)(y) = \sum_{k=1}^q b_k\big(a(x_1), \dots, a(x_m)\big) t_k(y) + c,$$

where $c \in \mathbb{R}$ is an optional bias term. For vector-valued outputs of dimension $n$, the latent dimension $q$ is partitioned as $0 = q_0 < q_1 < \cdots < q_n = q$, and each component is computed as

$$G_\theta^{(i)}(a)(y) = \sum_{k=q_{i-1}+1}^{q_i} b_k\big(a(x_1), \dots, a(x_m)\big) t_k(y) + c, \quad i = 1, \dots, n.$$

This formulation enables DeepONets to flexibly approximate both scalar- and vector-valued operators.

To quantify the discrepancy between the DeepONet prediction and the true PDE solution, we define the pointwise mean-squared error for a given input $a$ as

$$\mathcal{L}(a, \theta) = \frac{1}{P} \sum_{j=1}^P \left| G_\theta(a)(y_j) - s(y_j) \right|^2,$$

where $\{y_j\}_{j=1}^P$ are evaluation locations in the domain of $G(a)$, and $s(y_j)$ denotes the corresponding PDE solution values obtained from the reference model. The total training objective is then obtained by averaging over all $N$ samples in the dataset:

$$\mathcal{L}(\theta) = \frac{1}{N} \sum_{i=1}^N \mathcal{L}\big(a^{(i)}, \theta\big).$$

### 3.4 Fourier Neural Operators

Fourier Neural Operators (FNOs) are another class of operator networks. The central building block of an FNO is the Fourier layer, which replaces traditional local convolutions with multiplications in the frequency domain. Given an intermediate feature representation $v(x)$ defined on a discretized domain, the Fourier layer computes

$$v'(x) \;=\; \mathcal{F}^{-1}\big(R_\theta(\hat{v}(\xi))\big)(x) \;+\; Wv(x), \tag{6}$$

where $\hat{v}(\xi) = \mathcal{F}[v](\xi)$ is the Fourier transform of $v$, $R_\theta$ denotes a learnable spectral multiplier acting on a truncated set of modes, and $W$ is a trainable pointwise linear transformation. The inverse Fourier transform $\mathcal{F}^{-1}$ maps the modified representation back to the spatial domain. An FNO consists of a sequence of Fourier layers interleaved with pointwise nonlinearities. The network is typically structured as

$$v^{(0)} = P(a), \quad v^{(\ell+1)} = \sigma\Big(\mathcal{F}^{-1}\big(R_\theta(\hat{v}^{(\ell)})\big) + Wv^{(\ell)}\Big),$$

where $P$ is a lifting operator that maps the input $a$ into a higher-dimensional latent space, and $\sigma$ is a nonlinearity such as tanh or ReLU. A final projection operator $Q$ maps the latent representation back to the output space:

$$\mathcal{G}_\theta(a) = Q(v^{(L)}).$$

By operating directly in the Fourier domain, FNOs efficiently capture long-range dependencies and global interactions, which are challenging for purely local architectures. However, standard FNOs rely on globally supported Fourier modes and lack explicit mechanisms for localized or adaptive refinement, which can degrade performance near sharp gradients or localized features.

**Unified perspective.** Despite their different training settings, PINNs, DeepONets, and FNOs share a common limitation: they rely primarily on global parameterizations. This can hinder accuracy when the target solution or operator contains fine-scale or spatially localized structure. Partition-of-unity models address this issue by blending local components while preserving global continuity. The approach developed here extends that idea by learning both the partitions and the hierarchical refinement across scales, combining geometry-aware gating with residual multi-level specialization.

### 3.5 Partition-of-Unity and Mixture-of-Experts Formulations

Partition-of-unity (POU) methods provide a general mathematical framework for constructing globally continuous approximations from a collection of local approximants. Originally developed in the context of finite element and mesh-free methods , the POU principle ensures global smoothness by blending overlapping local basis functions through nonnegative weighting functions that sum to one over the domain. Formally, let $\{\phi_j(\mathbf{x})\}_{j=1}^{N_p}$ denote a set of partition functions satisfying

$$\phi_j(\mathbf{x}) \geq 0, \qquad \text{and} \qquad \sum_{j=1}^{N_p} \phi_j(\mathbf{x}) = 1, \quad \forall \mathbf{x} \in \Omega. \tag{7}$$

Each $\phi_j$ is supported primarily on a subregion $\Omega_j \subset \Omega$, with overlapping supports across neighboring partitions.

Mixture-of-Experts (MoE) models provide a closely related perspective by representing a global function as a weighted combination of local approximants. Given a collection of expert functions $\{f_j(\mathbf{x})\}_{j=1}^{N_c}$, the global approximation is expressed as

$$u(\mathbf{x}) = \sum_{j=1}^{N_c} \phi_j(\mathbf{x})\, f_j(\mathbf{x}), \tag{8}$$

where the weighting functions $\phi_j$ act as a gating mechanism. When the weights satisfy the partition-of-unity conditions (7), the resulting model inherits the stability and smoothness properties of classical POU formulations.

In this context, the POU framework can be interpreted as a structured MoE model in which the gating functions are explicitly constrained to be nonnegative and to sum to one. The experts $\{f_j\}$ may be linear models, radial basis functions, multilayer perceptrons, or other specialized subnetworks. When the experts are nonlinear subnetworks, the model becomes a locally adaptive ensemble in which each expert specializes in a region of the input space, while smooth blending across regions is guaranteed by the partition-of-unity property.

In practice, the gating functions $\{\phi_j(\mathbf{x})\}$ may be parameterized through normalized radial basis functions, distance-based kernels, or other geometry-aware constructions. In this work, we define the gating functions using learnable, spatially structured mechanisms, which combine RBF activations with box initialization-based partitions—which are described in detail in Section 4. This design preserves the interpretability and continuity guarantees of classical POU methods while enabling data-driven, adaptive partitioning within a neural network framework.

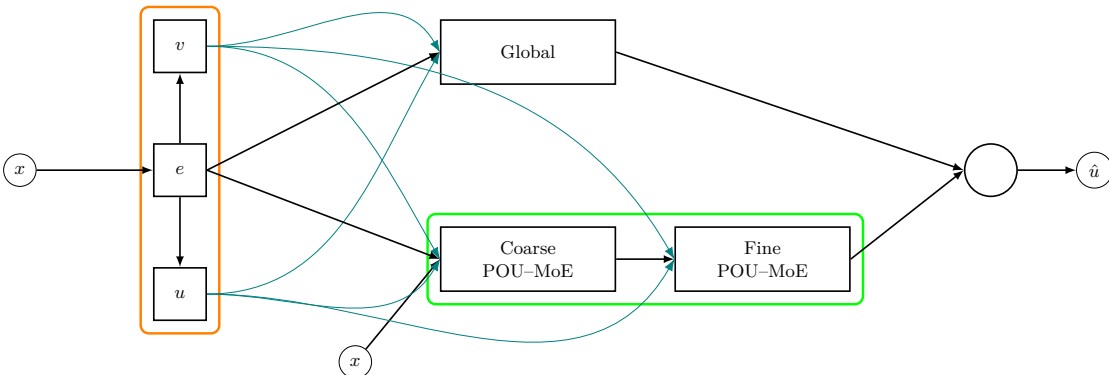

Figure 1: Hierarchical residual architecture used in HiRefPOU. The input $x$ is embedded and passed through a global backbone to obtain $u_\mathrm{g}(x)$. A coarse Mixture-of-Experts (MoE) layer predicts $u_c(x)$. A second MoE layer further predicts the fine-scale contributions $u_f(x)$ via parent-to-child refinement. The final output is the sum of all three components, $u(x) = u_\mathrm{g}(x) + u_c(x) + u_f(x)$.

## 4    Hierarchical Partition-of-Unity Models for Adaptive Operator Learning

We introduce HiRefPOU, a hierarchical refinement architecture that combines a global neural approximator with multi-level Partition-of-Unity Mixture-of-Experts (POU–MoE) subnetworks. The core idea is to decompose the target solution into a global component that captures smooth structure followed by localized networks that resolve intermediate and fine-scale features. While the framework can be instantiated in physics-informed settings, we emphasize that the models developed and evaluated in this section are primarily designed to address limitations in neural operator learning through adaptive, learned domain decomposition. In this work, we investigate one-, two-, and three-level instantiations, with the two-level architecture serving as the primary hierarchical formulation and the three-level variant used to probe deeper refinement. The construction extends naturally to an arbitrary number of levels.The resulting model yields a hierarchical residual decomposition, in which each level concentrates on the portion of the solution not resolved by the preceding levels. This results in a multiresolution approximation suitable to heterogeneous or multiscale PDE and operator-learning problems.

### 4.1    Hierarchical Residual POU Network

Formally, given an input $x \in \mathbb{R}^d$, the model produces the prediction

$$u_\theta(x) = u_\mathrm{g}(x;\theta_g) + u_c(x;\theta_c) + u_f(x;\theta_f), \tag{9}$$

where $u_\text{g}$ is a global baseline, $u_c$ captures localized medium-scale corrections, and $u_f$ provides highly localized refinements. The refinement is explicitly residual in that each level provides an additive correction that refines the approximation produced by previous levels, forming a multiscale expansion. Figure 1 shows our full network architecture diagram.

### 4.2 Global Approximator Network.

The global approximator $(u_g(x; \theta_g))$ is implemented as a modified residual network that incorporates shared conditioning features to stabilize optimization and capture smooth, low-frequency structure (Wang et al., 2021a; 2024; Rodriguez et al., 2024). Let $e(x) \in \mathbb{R}^m$ denote the shared embedding of the input coordinate. Starting from $g^{(0)}(x) = e(x)$, the global network applies a sequence of $L$ gated bottleneck blocks of the form

$$g^{(\ell+1)}(x) = \mathcal{B}\Big(g^{(\ell)}(x),\, h_u(x),\, h_v(x)\Big), \qquad \ell = 0, \dots, L-1, \tag{10}$$

where $h_u(x)$ and $h_v(x)$ are shared conditioning vectors derived from the embedding. Each bottleneck block consists of a stack of fully connected layers interleaved with nonlinear activations, followed by elementwise gating using the conditioning features. Specifically, intermediate activations are modulated according to

$$z \;\mapsto\; z \odot h_u(x) + \big(1 - z\big) \odot h_v(x), \tag{11}$$

which enables the network to interpolate between two learned feature representations and improves stability across heterogeneous regions of the domain (Wang et al., 2021a).

To promote stable training and encourage the global network to first learn a coarse approximation, each block is equipped with a learnable residual connection. Let $x_\text{in}$ denote the block input and $x_\text{out}$ its transformed output. The final block output is given by

$$\mathcal{B}(x_\text{in}, h_u, h_v) = \alpha\, x_\text{out} + (1 - \alpha)\, x_\text{in}, \tag{12}$$

where $\alpha$ is a scalar parameter initialized near zero and learned during training. This design allows the network to begin close to the identity map and gradually increase expressivity as optimization proceeds (Cooley et al., 2025; Wang et al., 2024). A final linear projection produces the global prediction

$$u_\text{g}(x) = W_\text{g}\, g^{(L)}(x). \tag{13}$$

The resulting global component provides a smooth baseline approximation that captures large-scale solution structure and serves as a stable reference for subsequent localized refinements.

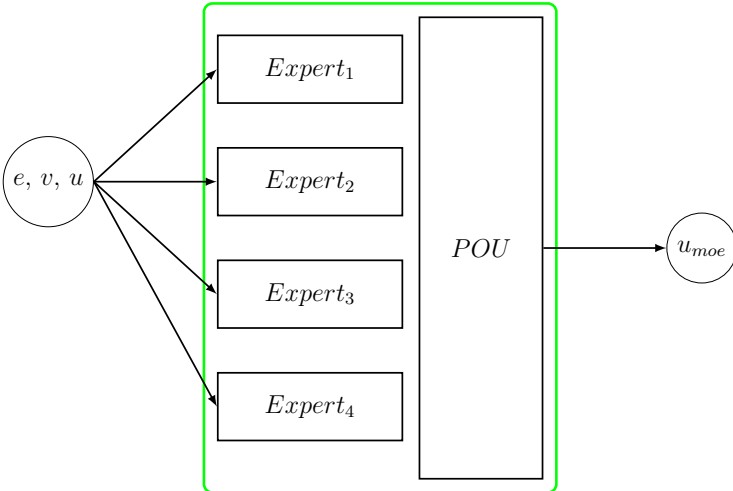

Figure 2: A POU-Mixture-of-Experts (MoE) layer predicts $u_p(x)$.

### 4.3 Geometry-Aware Gating Networks for Adaptive Domain Decomposition

A central contribution of this work is a geometry-aware gating network that combines center-based distance partitions with box-initialized residual gating. This design generalizes classical radial basis function (RBF) partitions while retaining the flexibility of learned, anisotropic domain decompositions. Each partition is associated with a learnable center point $c_j \in \mathbb{R}^d$, which acts as an explicit geometric anchor, allowing for easier partition initialization, regularization, and interpretation. We then compute the Euclidean distances as,

$$d_j^2(x) = \|x - c_j\|^2, \tag{14}$$

which serve as intrinsic geometric features that define the proximity of each point to the available partitions. These distances help define the center-based gating logits,

$$z_j^{\mathrm{cntr}}(x) = -\frac{d_j^2(x)}{2\alpha_j^2}, \tag{15}$$

where $\alpha_j > 0$ is a learnable, per-center scaling parameter that controls the sharpness of the partition. This formulation preserves the interpretability and smoothness of RBF-like partitions while allowing the effective support of each region to adapt during training.

To further increase the flexibility of the isotropic radial supports, we combine the center-based gating with a box-initialized residual gating network. This network consists of a shallow multilayer perceptron with residual connections, initialized such that its decision boundaries form approximately axis-aligned, non-overlapping hyperrectangular regions across the domain. During training, these initial box-like partitions are progressively deformed through learned nonlinear transformations, enabling the gating to capture anisotropic and nonconvex structures.

Let $z^{\mathrm{box}}(x)$ denote the logits produced by the box-based gating network and $z^{\mathrm{cntr}}(x)$ those produced by the center-distance gate. The final gating logits are obtained through a learnable convex combination

$$z(x) = w\, z^{\mathrm{box}}(x) + (1 - w)\, z^{\mathrm{cntr}}(x), \tag{16}$$

where $w = \mathrm{sigmoid}(m)$ is a scalar mixing coefficient obtained by applying a sigmoid to a trainable parameter $m$ such that $w \in (0, 1)$. This adaptive interpolation allows the model to transition smoothly between geometry-dominated and fully learned gating regimes.

Finally, a softmax is applied to the combined logits to obtain smooth POU weights,

$$\phi_j(x) = \frac{\exp(z_j(x))}{\sum_k \exp(z_k(x))}, \tag{17}$$

which satisfy $\sum_j \phi_j(x) = 1$ for all $x$. These weights define continuous domain partitions that are jointly optimized with the expert networks during training.

We emphasize that the gating construction above is written in the coordinate representation provided to the model. In Euclidean domains, this corresponds directly to the physical coordinates. For the three-dimensional reaction–diffusion benchmark considered here, the domain is the ball, not a manifold-valued surface. Thus, the gating network is applied directly to the Euclidean coordinates used by the operator model, and we do not introduce any intrinsic surface parameterization, geodesic distance, or manifold-specific gating constraints. Any locality is therefore learned in this ambient coordinate representation.

### 4.4 Partition-of-Unity Mixture-of-Experts Ensembles

Each POU level ($u_c$ and $u_f$ in Equation (9)) is implemented as a MOE ensemble composed of independently parameterized residual multilayer perceptrons. Let $\{\phi_j(x)\}_{j=1}^{N_c}$ denote the partition weights produced by the gating network at a given level. The corresponding expert ensemble produces local expert outputs $\{f_j(x)\}_{j=1}^{N_c}$, which are combined through a weighted sum.

Similar to the global approximator network $(u_g)$, each expert $f_j$ is implemented as a residual modified MLP conditioned on the shared features $(h_u(x), h_v(x))$. Given an input feature vector $x_0$, the expert applies a sequence of fully connected layers,

$$x_{k+1} = \sigma(W_k x_k), \qquad k = 0, \ldots, D - 1, \tag{18}$$

where $\sigma$ denotes a nonlinear activation function. As in the global network, each intermediate activation is modulated through the shared conditioning,

$$x_k \ \mapsto \ x_k \odot h_u(x) + \big(1 - x_k\big) \odot h_v(x), \tag{19}$$

which enables coordinated feature sharing across experts while preserving independent parameters.

To improve optimization and prevent overfitting in early training, each expert incorporates a learnable residual connection. Let $x_{\mathrm{in}}$ denote the expert input and $x_{\mathrm{out}}$ the transformed output. The residual update is given by

$$\tilde{x} = \alpha\, x_{\mathrm{out}} + (1 - \alpha)\, x_{\mathrm{in}}, \tag{20}$$

where $\alpha$ is a learnable scalar initialized near zero. A final linear projection maps $\tilde{x}$ to a scalar expert output.

All experts at a given level are evaluated in parallel using vectorized execution with independent parameters. The ensemble output is then formed as,

$$u_{\mathrm{POU}}(x) = \sum_{j=1}^{N_c} \phi_j(x)\, f_j(x), \tag{21}$$

which guarantees smooth blending across partitions and preserves global continuity through the POU constraint. This design enables each expert to specialize to localized features within its assigned region while the residual structure and shared conditioning promote stable training and coherent behavior across the ensemble and global network.

## 4.5 Geometric Regularization for Learned Partitions

To encourage well-distributed domain partitions, we regularize the gating network and its learnable centers using lightweight geometric losses. These regularizers are applied either during a short pretraining phase of the gating networks or jointly with the full model, and do not require access to solution data.

Let $X = \{x_i\}_{i=1}^{N} \subset \Omega$ denote a set of sample points and $C = \{c_j\}_{j=1}^{N_p}$ the corresponding set of learnable partition centers. For $\Omega \subset \mathbb{R}^d$, we define a generic coverage regularizer by comparing a geometric measure of the spread of the centers to the measure of the domain. In one dimension, this reduces to the length of the interval spanned by the centers; in two dimensions, to the area of their convex hull; and in three dimensions, to the volume of their convex hull. Denoting this measure generically by $M(C)$ and the corresponding domain measure by $M_\Omega$, we write

$$L_{\mathrm{cov}} = \lambda_{\mathrm{cov}} \frac{|M(C) - M_\Omega|}{M_\Omega + \varepsilon}, \tag{22}$$

which discourages collapsed or poorly distributed center configurations.

For example, when $\Omega = [0, 1]$, we may take

$$M(C) = \max_j c_j - \min_j c_j, \qquad M_\Omega = 1. \tag{23}$$

When $\Omega = [0, 1]^2$, $M(C)$ is the area of the convex hull of the centers, recovering the two-dimensional form used in this work. When $\Omega = [0, 1]^3$, $M(C)$ is the volume of the convex hull of the centers.

We also apply a soft containment penalty to discourage centers from drifting outside the domain. For box domains $\Omega = [0, 1]^d$, this penalty extends coordinatewise as

$$L_{\mathrm{cont}} = \lambda_{\mathrm{cont}} \sum_{j=1}^{N_p} \sum_{k=1}^{d} \left[ \max(0, -c_{j,k}) + \max(0, c_{j,k} - 1) \right], \tag{24}$$

where $c_{j,k}$ denotes the $k$th coordinate of center $c_j$. In one dimension this reduces to the obvious two-sided interval penalty, while in three dimensions it adds the corresponding penalties over the three coordinate directions.

### 4.6 Hierarchical Partition-of-Unity Refinement

While a single-level POU network can effectively localize approximation tasks, its representational capacity is inherently limited by the resolution of the initial partitioning. To address this limitation, we introduce a hierarchical extension, denoted HiRefPOU-2L, which organizes multiple POU MoE blocks into a residual, multi-level refinement scheme. In this construction, each POU level contributes an additive correction that refines the solution produced by coarser levels, enabling progressively finer localization where needed.

Let $\{\phi_j^c(x)\}_{j=1}^{N_c}$ denote the coarse-level POU weights, produced by a gating network as described in the previous section. These weights satisfy

$$\sum_{j=1}^{N_c} \phi_j^c(x) = 1, \qquad \forall x \in \Omega.$$

The coarse approximation in Equation (9) is defined as

$$u_c(x) = \sum_{j=1}^{N_c} \phi_j^c(x)\, f_j^c(x), \tag{25}$$

where $f_j^c(x)$ denotes the expert associated with the $j$-th coarse partition.

To introduce hierarchical refinement, each coarse partition $\phi_j^c(x)$ is then subdivided by a second gating network that produces child weights $\{\psi_{j,k}(x)\}_{k=1}^{N_f}$, locally normalized within each parent partition. The resulting fine-level POU weights are defined by

$$\phi_{j,k}^f(x) = \phi_j^c(x)\, \psi_{j,k}(x), \qquad \sum_{k=1}^{N_f} \psi_{j,k}(x) = 1,$$

which guarantees the nested POU property $\sum_{j=1}^{N_c} \sum_{k=1}^{N_f} \phi_{j,k}^f(x) = 1$.

In practice, this constraint is satisfied exactly by defining each fine-level weight as the product of its parent coarse weight and a locally normalized softmax over the children of that parent. Each fine partition $\phi_{j,k}^f(x)$ is paired with its own local expert $f_{j,k}^f(x)$, yielding the fine-scale correction from Equation (9) as,

$$u_f(x) = \sum_{j=1}^{N_c} \sum_{k=1}^{N_f} \phi_{j,k}^f(x)\, f_{j,k}^f(x). \tag{26}$$

This structure mirrors the skip-connection paradigm of residual networks, with the key distinction that refinement is organized through nested POU weights rather than unstructured feature addition. The coarse approximant $u_c(x)$ captures mid-frequency structure, while the fine correction $u_f(x)$ enables localized specialization within each coarse region. Although we focus on a two-level hierarchy in this work, the construction extends naturally to deeper refinement trees by recursively subdividing partitions and composing additional residual POU correction terms. The full hierarchical POU model is trained end-to-end by minimizing the appropriate physics-informed or data-driven loss for the target application.

## 5 Neural Operator Instantiations of the POU Framework

The Partition-of-Unity (POU) framework extends naturally to neural operator architectures.

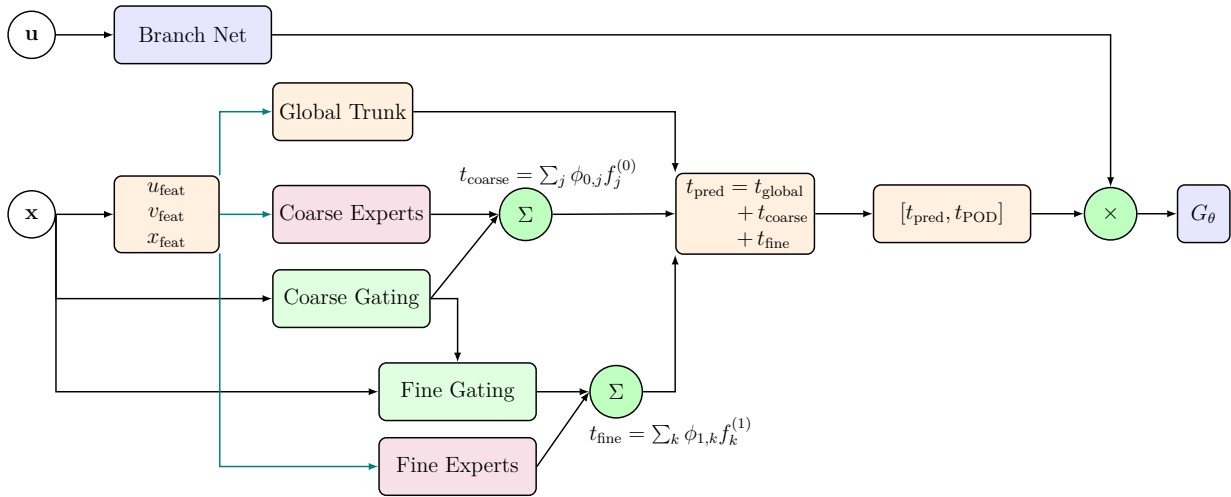

Figure 3: Hierarchical residual architecture used in HiRefPOU-DeepONet. The input $x$ is embedded and passed through a global backbone to obtain $t_g(x)$. A coarse Mixture-of-Experts (MoE) layer predicts $t_c(x)$. A second MoE layer further decomposes into fine-scale contributions $t_f(x)$ via parent-to-child refinement. The final output is the sum of all three components, $t(x) = t_g(x) + t_c(x) + t_f(x)$.

### 5.1 Hierarchical POU Deep Operator Networks (HiRefPOU-DeepONets)

We extend the POU framework to operator learning by introducing HiRefPOU-DeepONets, which incorporate spatially localized expert structure into the trunk representation of a Deep Operator Network (DeepONet) (Lu et al., 2021). As reviewed in Section 3, a standard DeepONet represents an operator through an inner product between (i) a branch network encoding an input function and (ii) a trunk network encoding evaluation coordinates.

The proposed construction mirrors the residual POU refinement strategy used in HiRefPOU for PDE approximation, but applies it instead to the coordinate-dependent component of the operator representation. In our formulation, the branch network remains global and maps sampled observations of the input function $a$ to a coefficient vector $b(a) \in \mathbb{R}^{p+p_{\text{pod}}}$. The trunk representation is augmented with a POU mixture-of-experts defined over the evaluation coordinate $y$.

Specifically, we define a global trunk feature $t_g(y) \in \mathbb{R}^p$ together with a collection of $N_c$ coarse local trunk experts $\{t_j(y) \in \mathbb{R}^p\}_{j=1}^{N_c}$. These experts are blended using smooth partition-of-unity weights $\{\phi_j(y)\}_{j=1}^{N_c}$ satisfying $\sum_{j=1}^{N_c} \phi_j(y) = 1$, yielding the POU-enhanced trunk feature

$$t_{\text{c}}(y) = t_{\text{g}}(y) + \sum_{j=1}^{N_c} \phi_j(y)\, t_j(y), \qquad t_{\text{c}}(y) \in \mathbb{R}^p. \tag{27}$$

To incorporate problem-specific low-rank structure, we concatenate $t_c(y)$ with a fixed POD basis vector $t_c(y) \in \mathbb{R}^{p_{\text{pod}}}$, forming the final trunk feature

$$t(y) = \big[t_{\text{c}}(y),\, t_{\text{pod}}(y)\big] \in \mathbb{R}^{p+p_{\text{pod}}}. \tag{28}$$

The operator prediction is then obtained via the standard DeepONet contraction

$$\hat{G}(a)(y) = \langle b(a),\, t(y) \rangle + b, \tag{29}$$

where $b \in \mathbb{R}$ is a learned bias. This formulation preserves the global expressivity of DeepONets while enabling spatially localized specialization through the POU-weighted trunk experts.

We further introduce a hierarchical two-level POU structure on the trunk side to enable multi-resolution operator representations. At the coarse level, a gating network produces weights $\{\phi_j^c(y)\}_{j=1}^{N_c}$ satisfying

$\sum_{j=1}^{N_c} \phi_j^c(y) = 1$, which blend a set of coarse trunk experts. Each coarse partition is then subdivided by a second gating network that produces child weights $\{\psi_{j,k}(y)\}_{k=1}^{N_f}$, locally normalized within each parent partition. The resulting fine-level POU weights are defined by

$$\phi_{j,k}^f(y) = \phi_j^c(y)\,\psi_{j,k}(y), \qquad \sum_{k=1}^{N_f} \psi_{j,k}(y) = 1,$$

which guarantees the nested POU property $\sum_{j=1}^{N_c}\sum_{k=1}^{N_f} \phi_{j,k}^f(y) = 1$. The full hierarchical POU trunk representation is therefore given by

$$t_{\text{pou}}(y) = t_{\text{g}}(y) + \sum_{j=1}^{N_c} \phi_j^c(y)\, t_j^c(y) + \sum_{j=1}^{N_c}\sum_{k=1}^{N_f} \phi_{j,k}^f(y)\, t_{j,k}^f(y), \tag{30}$$

where $t_j^c$ and $t_{j,k}^f$ denote coarse- and fine-level trunk experts, respectively. This hierarchical construction enables localized refinement of the operator representation while preserving global consistency through the POU structure.

## 5.2 Partition-of-Unity Fourier Neural Operators (POU-FNO)

In their standard form, FNOs act globally on the entire spatial domain, which can limit their ability to capture strongly localized features. We extend the partition-of-unity (POU) framework to FNOs by introducing a geometry-aware gating mechanism that blends multiple operator components using smooth, spatially localized weights. Importantly, the POU-FNO reuses the same geometry-aware gating network introduced in Section 4.3, with no changes to its functional form or training procedure. The resulting architecture combines global spectral operator learning with spatially localized POU weighting, enabling spatial adaptivity and localized feature learning. We emphasize that the POU-FNO is intended as a simple architectural extension illustrating the generality of the POU principle, rather than as a primary hierarchical contribution of this work.

For each spatial coordinate $y_\ell$, a shared gating network produces logits $\ell(y_\ell) \in \mathbb{R}^{N_p}$, which are converted into partition weights as described in Section 4.3. The gate depends only on spatial coordinates and is shared across the batch, ensuring geometry-consistent partitions. Because the POU weights are applied pointwise in physical space, this construction preserves the global FFT structure of the FNO and avoids introducing artificial subdomain boundaries.

A single 1D FNO layer updates hidden features $h(y) \in \mathbb{R}^C$ via

$$\tilde{h}(y) = \mathcal{F}^{-1}\Big( \sum_{|k|\leq K} \hat{W}_k\, \mathcal{F}[h]_k \Big) + W_1 h(y), \tag{31}$$

where $\mathcal{F}$ and $\mathcal{F}^{-1}$ denote the forward and inverse discrete Fourier transforms, $\hat{W}_k \in \mathbb{C}^{C\times C}$ are learnable complex-valued spectral coefficients applied to the lowest $K$ modes, and $W_1 \in \mathbb{R}^{C\times C}$ is a learned pointwise linear transformation. Equation (31) is followed by a nonlinearity such as ReLU or GELU, and stacked for $D$ layers to form an FNO block, which may be either shared across partitions or instantiated per partition depending on the architectural variant.

Given $N_p$ expert predictions $\{\hat{v}_p(y)\}_{p=1}^{N_p}$, the final output is formed by a POU-weighted sum,

$$\hat{v}(y_\ell) = \sum_{p=1}^{N_p} \phi_p(y_\ell)\, \hat{v}_p(y_\ell), \tag{32}$$

which ensures smooth transitions between partitions. For $N_p = 1$, the model reduces exactly to a standard global FNO.

We consider two variations of this POU-FNO construction. In the first (POU-ExpertFNO), each partition is associated with an independent local FNO, resulting in a fully localized spectral mixture. In the second

(POU-SharedFNO), which we use for most experiments, a single global FNO backbone is shared across all partitions, and locality is introduced through partition-specific expert heads applied to the shared spectral features. Both variants satisfy the POU aggregation in (32) and differ only in the degree to which spectral parameters are shared.

### 5.3 Computational Complexity of HiRefPOU-DeepONets

We summarize the computational complexity of the HiRefPOU-DeepONet architectures in terms of the branch, trunk, expert, and gating components. Let $B$ denote the batch size and $N_y$ the number of evaluation coordinates per input function. Let $p$ denote the learned trunk latent dimension and $p_{\text{pod}}$ the number of appended POD modes. We assume that the global trunk network and each local trunk expert have comparable hidden width $w_t$ and depth $L_t$, while the gating networks have hidden width $w_g$ and depth $L_g$. We denote the cost of evaluating the branch network on a batch of $B$ input functions by $C_{\text{branch}}(B)$.

Because the trunk and gating networks depend only on the evaluation coordinate $y$, their cost depends on whether all batch elements share the same evaluation grid. We define

$$\eta_y = \begin{cases} N_y, & \text{if all batch elements share the same evaluation coordinates,} \\ BN_y, & \text{if evaluation coordinates differ across batch elements.} \end{cases}$$

The final DeepONet contraction is evaluated for every sample-coordinate pair and therefore scales as

$$\mathcal{O}(BN_y(p + p_{\text{pod}})).$$

The fixed POD basis does not introduce trainable parameters and contributes only through this final concatenation and contraction.

For HiRefPOU-1L with $M$ local trunk experts, the trunk representation is

$$t_{\text{pou}}(y) = t_g(y) + \sum_{j=1}^{M} \phi_j(y) t_j(y).$$

Under the current dense implementation, all experts are evaluated at all query coordinates. The dominant forward-pass cost is therefore

$$C_{1L} = \mathcal{O}\big(C_{\text{branch}}(B) + \eta_y(1 + M)L_t w_t^2 + C_\phi(M) + BN_y(p + p_{\text{pod}})\big),$$

where $C_\phi(M)$ is the cost of evaluating the gate with $M$ outputs. For a dense MLP gate,

$$C_\phi(M) = \mathcal{O}\big(\eta_y(L_g w_g^2 + w_g M + Md)\big),$$

where the $Md$ term accounts for center-distance features in $d$ spatial dimensions.

For HiRefPOU-2L, let $M_c$ denote the number of coarse experts and $M_f$ the total number of fine experts across all coarse parents. The hierarchical trunk representation contains $1 + M_c + M_f$ densely evaluated trunk networks, so the dominant forward-pass cost is

$$C_{2L} = \mathcal{O}\big(C_{\text{branch}}(B) + \eta_y(1 + M_c + M_f)L_t w_t^2 + C_{\phi_c}(M_c) + C_\psi(M_f) + BN_y(p + p_{\text{pod}})\big).$$

If each coarse parent has $K_f$ children, then $M_f = M_c K_f$, recovering the balanced-tree scaling.

Ignoring lower-order input and output projections, the corresponding trainable parameter counts scale as

$$P_{1L} = \mathcal{O}\big(P_{\text{branch}} + (1 + M)L_t w_t^2 + L_g w_g^2 + w_g M\big),$$

and

$$P_{2L} = \mathcal{O}\big(P_{\text{branch}} + (1 + M_c + M_f)L_t w_t^2 + 2L_g w_g^2 + w_g M_c + w_g M_f\big).$$

More generally, for a hierarchy with $L_h$ POU refinement levels and $M_\ell$ experts at level $\ell$, the dense expert-evaluation cost scales linearly with the total number of experts:

$$C_{\text{trunk}} = \mathcal{O}\left(\eta_y \left(1 + \sum_{\ell=1}^{L_h} M_\ell\right) L_t w_t^2\right).$$

Thus, the practical cost of the current implementation grows with the total number of trunk experts across levels. Since all experts are evaluated densely at all query coordinates, the current implementation does not obtain computational savings from sparse routing. An ideal routed implementation could reduce the constant factor by evaluating only the active experts associated with each coordinate, but this is not used in the experiments reported here.

## 6 Benchmark problems and implementation details

### 6.1 Baseline Comparison Methods

To evaluate the proposed HiRefPOU framework, we primarily benchmark against established neural operator (NO) architectures. Our focus is on operator learning problems, where the objective is to approximate mappings between infinite-dimensional function spaces rather than individual PDE solutions. Accordingly, the majority of our numerical experiments compare HiRefPOU to representative operator-learning baselines under matched training protocols and, where possible, comparable DeepONet-family partition or parameter budgets.

**Neural Operator Baselines.** We compare against the following operator-learning methods:

- **Vanilla DeepONet**: the standard DeepONet architecture with a single global trunk network and no explicit spatial localization.

- **(P+1)-Vanilla DeepONet**: an overparameterized baseline consisting of $P + 1$ independent global trunk networks, where $P$ matches the number of partitions used in POU-based models. This baseline tests whether performance gains arise purely from increased parameter count rather than spatial localization.

- **(POD)**: a DeepONet augmented with a fixed Proper Orthogonal Decomposition (POD) basis in the trunk, providing a global low-rank representation of the solution space.

- **(Vanilla, POU)** (Sharma & Shankar, 2025): a DeepONet with a partition-of-unity trunk using fixed (non-adaptive) spatial partitions, isolating the effect of localization without adaptive gating.

- **(POD, POU)** (Sharma & Shankar, 2025): a localized DeepONet combining global POD modes with fixed POU-based spatial experts.

- **(Vanilla, POD, POU)** (Sharma & Shankar, 2025): a hybrid DeepONet that includes global vanilla, global POD, and localized POU expert components, testing whether additional global capacity improves upon POD–POU alone.

- **Geo-Fourier Neural Operator (GeoFNO)** (Li et al., 2023): which extends the Fourier Neural Operator to irregular geometries and performs global spectral convolution in Fourier space. GeoFNO is particularly well-suited for learning solution operators of PDEs posed on complex domains.

- **LocalFNO** (Liu-Schiaffini et al., 2024): a Fourier neural operator augmented with localized integral and/or differential kernel layers. Unlike standard FNOs, which rely entirely on global spectral convolutions, this architecture introduces explicit local receptive fields. We include LocalFNO as a strong locality-aware spectral baseline for comparison with our POU-based localized operator models.

These models provide strong baselines for assessing generalization accuracy, scalability, and computational efficiency in operator-learning settings.

**Motivating Physics-Informed Case Study.** In addition to operator-learning benchmarks, we include a limited set of physics-informed neural network (PINN) experiments as a motivating case study. Specifically, we consider a two-dimensional Darcy flow problem with discontinuous coefficients, which highlights known limitations of globally smooth neural approximations when applied to interface-dominated PDEs. These experiments are not intended as an exhaustive comparison with PINN or domain-decomposition variants such as xPINNs, but rather to illustrate why localized representations are advantageous for problems with sharp material interfaces. For completeness, we compare against representative PINN-based methods in this setting, including: the standard PINN formulation (Raissi, 2018), Fourier-feature PINNs (Wang et al., 2021b), and PirateNets (Wang et al., 2024). Additional PINN experiments and implementation details are provided in Appendix A.

## 6.2 Training Setup and Evaluation Metrics

To assess predictive accuracy, we report the relative $\ell_2$ error, which quantifies the discrepancy between the network prediction and the ground truth solution. Specifically, the error is defined as

$$\ell_2 = \frac{\sqrt{\sum_{i=1}^{N} |u(x_i, t_i) - \hat{u}(x_i, t_i)|^2}}{\sqrt{\sum_{i=1}^{N} |u(x_i, t_i)|^2}}, \tag{33}$$

where $u$ denotes the exact solution and $\hat{u}$ corresponds to the approximation produced by the neural network. This metric provides a relative measure of the error by normalizing with respect to the magnitude of the reference solution. For the operator-learning benchmarks, we report relative $\ell_2$ and, in the detailed tables, $\ell_\infty$ errors together with training and inference times. For the physics-informed Darcy case study, residual and boundary losses are used during training, while the reported metrics focus on solution and flux errors.

Unless otherwise specified, all models were trained using the Adam optimizer (Kingma & Ba, 2015) with an inverse exponential-annealing learning rate decay. Each training run was performed for 150,000 iterations. Each reported result represents the mean over five random seeds. Training was conducted on an NVIDIA H100 GPU under JAX/Flax with single-precision arithmetic. All code was implemented in Python 3.10.18 using JAX 0.6.2, with DeepMind Haiku 0.0.14 for neural network modules and Optax 0.2.5 for optimization[1]. Additional architecture and hyperparameter details for each model and experiment are summarized in Appendix A.

## 6.3 HiRefPOU Model Variants and Implementation Details

For each problem, we compare the following variants of the proposed hierarchical POU architecture, each corresponding to a different degree of gating adaptivity during training:

- **HiRefPOU-1L**: a single-level POU model $u(x) = u_g(x) + u_0(x)$ in which the coarse gating network is co-trained jointly with the global network and local experts. This allows the partition to continuously adapt to the evolving residuals.

- **HiRefPOU-2L**: the full hierarchical POU model in which the coarse and fine gates are co-trained along all other model parameters throughout PDE/operator training. This configuration allows both partition levels to continuously adapt to the evolving multiscale residual structure.

- **HiRefPOU-3L**: a deeper hierarchical extension that adds an additional residual PoU refinement level. This variant is used to test whether deeper coarse-to-fine partitioning provides additional accuracy beyond the 2L hierarchy, while preserving the same overall residual PoU design.

In all cases, the training minimizes the combined PDE/operator loss together with the hierarchical residual losses described in Section 4.

---

[1]https://github.com/madicooley/AdaptiveOL

Further, to demonstrate the generality of the proposed POU framework, we additionally embed the POU mechanism within neural operator architectures as described in Section 5.2. These extensions are not hierarchical, but instead aim to illustrate how POU-based localization can improve operator learning in FNO models. Specifically, we consider the following POU-FNO realizations:

- **POU-SharedFNO**: a FNO with a shared global spectral backbone and partition-specific expert heads, offering a computationally efficient realization of POU-based localization.

- **POU-ExpertFNO**: a more expressive but costly variant in which each partition is associated with an independent local FNO, yielding a fully localized spectral mixture.

## 7 Numerical Results

Table 1: (2D Rough Darcy Flow) Solution and flux errors for the Darcy flow benchmark using HiRefPOU-1L/-2L, APINN, and PirateNet.

| Model | $(N_c, N_f)$ | Rel. $\ell_2$ $u$ Error | Rel. $\ell_2$ $flux-x$ Error | Abs. Error $flux-y$ |
|---|---|---|---|---|
| PirateNet | (-, -) | $5.1 \times 10^{-5} \pm 2.9 \times 10^{-5}$ | $2.8 \times 10^{-5} \pm 9.0 \times 10^{-6}$ | $8.0 \times 10^{-2} \pm 5.5 \times 10^{-2}$ |
| APINN | (2, -) | $3.7 \times 10^{-5} \pm 1.8 \times 10^{-5}$ | $2.7 \times 10^{-5} \pm 1.4 \times 10^{-5}$ | $4.1 \times 10^{-2} \pm 1.9 \times 10^{-2}$ |
| HiRefPOU-1L | (2, -) | $1.5 \times 10^{-7} \pm 4.6 \times 10^{-8}$ | $1.6 \times 10^{-7} \pm 2.9 \times 10^{-8}$ | $1.3 \times 10^{-4} \pm 1.2 \times 10^{-5}$ |
| HiRefPOU-2L | (2, 3) | $1.8 \times 10^{-7} \pm 3.8 \times 10^{-7}$ | $1.2 \times 10^{-7} \pm 1.9 \times 10^{-7}$ | $1.4 \times 10^{-4} \pm 2.2 \times 10^{-2}$ |

### 7.1 Motivating PINN Example: Darcy Flow with Discontinuous Coefficients

We begin with a two-dimensional Darcy flow problem with discontinuous coefficients, included as a motivating example to illustrate the limitations of globally smooth PINNs on interface-dominated PDEs. Darcy flow serves as a canonical elliptic benchmark in heterogeneous media, where accurate flux recovery across material interfaces is essential. The Darcy experiment is included as a motivating physics-informed case study, while the remaining experiments evaluate the proposed localization mechanisms in the operator-learning setting.

We consider the steady Darcy problem

$$-\nabla \cdot \big(\mu(x)\,\nabla\phi(x)\big) = f(x), \qquad x \in \Omega = [0,1]^2, \tag{34}$$

with mixed Dirichlet–Neumann boundary conditions. The permeability field $\mu(x)$ is piecewise constant and discontinuous, inducing sharp material interfaces.

**Five-strip benchmark.** We employ the standard five-strip manufactured-solution test (Trask et al., 2017; Nakshatrala et al., 2006), in which the domain is partitioned into horizontal layers with distinct permeability values. The exact solution is chosen such that the pressure and normal flux remain continuous across interfaces, while tangential flux components exhibit discontinuities.

Figure 14 and Table 1 compare solution and flux predictions for representative PINN-based methods and the proposed HiRefPOU. While standard PINN variants exhibit degraded accuracy near material interfaces, HiRefPOU-1L achieves near machine-precision accuracy in both the solution and flux components. This example highlights the benefit of localized, partition-based representations for PDEs with discontinuous coefficients. Further technical details, interface conditions, and additional PINN results are provided in Appendix A.

### 7.2 2D Lid-Driven Cavity Flow

*Problem Setup.* We next consider the two-dimensional lid-driven cavity flow problem, a classical benchmark for incompressible fluid dynamics in a confined domain. The flow is governed by the incompressible Navier–Stokes

Table 2: Testing relative $l_2$ errors for the **Burgers**, **cavity flow**, **Full-Field Burgers**, **Kuramoto–Sivashinsky**, **2D reaction–diffusion**, and **3D reaction–diffusion** problems. RD stands for reaction–diffusion. For each problem, the best result across all models is shown in **bold**, and the second-best result is underlined. Rows corresponding to the proposed adaptive PoU-based models are highlighted.

| | Model | Burgers | Cavity flow |
|---|---|---|---|
| **DeepONets** | Vanilla | $2.24 \times 10^{-1} \pm 3.0 \times 10^{-3}$ | $4.86 \times 10^{-2} \pm 6.2 \times 10^{-3}$ |
| | (P+1)-Vanilla | $2.25 \times 10^{-1} \pm 6.5 \times 10^{-3}$ | $2.05 \times 10^{-2} \pm 3.3 \times 10^{-3}$ |
| | (POD, PoU) | $1.95 \times 10^{-1} \pm 1.4 \times 10^{-3}$ | $2.28 \times 10^{-3} \pm 1.5 \times 10^{-4}$ |
| | (Vanilla, PoU) | $2.43 \times 10^{-1} \pm 1.1 \times 10^{-2}$ | $1.73 \times 10^{-2} \pm 4.7 \times 10^{-3}$ |
| | (Vanilla, POD, PoU) | $1.92 \times 10^{-1} \pm 2.4 \times 10^{-3}$ | $2.16 \times 10^{-3} \pm 1.7 \times 10^{-4}$ |
| | HiRefPOU-1L (POD) | $1.75 \times 10^{-1} \pm 1.1 \times 10^{-3}$ | $\mathbf{1.55 \times 10^{-3} \pm 1.3 \times 10^{-4}}$ |
| | HiRefPOU-2L (POD) | $1.74 \times 10^{-1} \pm 1.2 \times 10^{-3}$ | $\underline{1.69 \times 10^{-3} \pm 1.2 \times 10^{-4}}$ |
| | HiRefPOU-3L (POD) | $1.74 \times 10^{-1} \pm 2.1 \times 10^{-3}$ | $1.79 \times 10^{-3} \pm 9.6 \times 10^{-5}$ |
| **FNOs** | GeoFNO | $3.56 \times 10^{-1} \pm 3.0 \times 10^{-2}$ | $5.56 \times 10^{-2} \pm 1.2 \times 10^{-2}$ |
| | LocalFNO | $\mathbf{1.16 \times 10^{-1} \pm 4.4 \times 10^{-2}}$ | $1.38 \times 10^{-2} \pm 1.2 \times 10^{-3}$ |
| | POU-SharedFNO | $1.74 \times 10^{-1} \pm 3.7 \times 10^{-2}$ | $4.15 \times 10^{-2} \pm 1.5 \times 10^{-2}$ |
| | POU-ExpertFNO | $\underline{1.24 \times 10^{-1} \pm 1.3 \times 10^{-2}}$ | $8.07 \times 10^{-3} \pm 4.4 \times 10^{-5}$ |

| | Model | Full-Field Burgers | Kuramoto-Sivashinsky |
|---|---|---|---|
| **DeepONets** | Vanilla | $1.02 \times 10^{-1} \pm 2.0 \times 10^{-3}$ | $1.70 \pm 1.3 \times 10^{-1}$ |
| | (P+1)-Vanilla | $1.27 \times 10^{-1} \pm 2.8 \times 10^{-3}$ | $1.68 \pm 1.3 \times 10^{-1}$ |
| | (POD, PoU) | $8.84 \times 10^{-2} \pm 2.3 \times 10^{-3}$ | $1.59 \pm 2.2 \times 10^{-2}$ |
| | (Vanilla, PoU) | $1.01 \times 10^{-1} \pm 1.6 \times 10^{-3}$ | $1.58 \pm 1.2 \times 10^{-2}$ |
| | (Vanilla, POD, PoU) | $9.06 \times 10^{-2} \pm 1.0 \times 10^{-3}$ | $1.50 \pm 1.4 \times 10^{-2}$ |
| | HiRefPOU-1L (POD) | $8.76 \times 10^{-2} \pm 1.5 \times 10^{-3}$ | $1.25 \pm 6.3 \times 10^{-2}$ |
| | HiRefPOU-2L (POD) | $8.62 \times 10^{-2} \pm 2.2 \times 10^{-3}$ | $1.33 \pm 2.3 \times 10^{-2}$ |
| | HiRefPOU-3L (POD) | $8.68 \times 10^{-2} \pm 1.1 \times 10^{-3}$ | $1.39 \pm 2.2 \times 10^{-2}$ |
| **FNOs** | GeoFNO | $4.21 \times 10^{-2} \pm 3.5 \times 10^{-3}$ | $9.03 \times 10^{-1} \pm 4.5 \times 10^{-3}$ |
| | LocalFNO | $3.17 \times 10^{-2} \pm 4.3 \times 10^{-3}$ | $7.85 \times 10^{-1} \pm 9.9 \times 10^{-3}$ |
| | POU-SharedFNO | $\mathbf{2.49 \times 10^{-2} \pm 3.4 \times 10^{-3}}$ | $\mathbf{7.62 \times 10^{-1} \pm 1.5 \times 10^{-2}}$ |
| | POU-ExpertFNO | $\underline{2.73 \times 10^{-2} \pm 2.3 \times 10^{-3}}$ | $\underline{7.80 \times 10^{-1} \pm 1.9 \times 10^{-2}}$ |

| | Model | 2D RD | 3D RD |
|---|---|---|---|
| **DeepONets** | Vanilla | $1.61 \times 10^{-3} \pm 4.4 \times 10^{-4}$ | $1.20 \times 10^{-3} \pm 3.6 \times 10^{-4}$ |
| | (P+1)-Vanilla | $5.77 \times 10^{-4} \pm 2.8 \times 10^{-4}$ | $1.30 \times 10^{-1} \pm 1.4 \times 10^{-1}$ |
| | (POD, PoU) | $4.74 \times 10^{-4} \pm 9.8 \times 10^{-5}$ | $4.46 \times 10^{-4} \pm 2.0 \times 10^{-4}$ |
| | (Vanilla, PoU) | $7.81 \times 10^{-4} \pm 2.1 \times 10^{-4}$ | $5.57 \times 10^{-4} \pm 3.1 \times 10^{-4}$ |
| | (Vanilla, POD, PoU) | $7.74 \times 10^{-4} \pm 1.4 \times 10^{-4}$ | $4.22 \times 10^{-4} \pm 1.6 \times 10^{-4}$ |
| | HiRefPOU-1L (POD) | $4.48 \times 10^{-4} \pm 2.4 \times 10^{-4}$ | $3.68 \times 10^{-4} \pm 2.2 \times 10^{-4}$ |
| | HiRefPOU-2L (POD) | $\underline{3.68 \times 10^{-4} \pm 1.3 \times 10^{-4}}$ | $\underline{3.12 \times 10^{-4} \pm 1.6 \times 10^{-4}}$ |
| | HiRefPOU-3L (POD) | $\mathbf{3.56 \times 10^{-4} \pm 6.2 \times 10^{-5}}$ | $\mathbf{2.55 \times 10^{-4} \pm 3.0 \times 10^{-5}}$ |
| **FNOs** | GeoFNO | $4.84 \times 10^{-2} \pm 1.1 \times 10^{-1}$ | $8.56 \times 10^{-1} \pm 5.2 \times 10^{-1}$ |
| | LocalFNO | $6.31 \times 10^{-3} \pm 4.9 \times 10^{-6}$ | $9.30 \times 10^{-3} \pm 2.3 \times 10^{-5}$ |
| | POU-SharedFNO | $7.47 \times 10^{-3} \pm 2.1 \times 10^{-4}$ | $1.02 \pm 1.2$ |
| | POU-ExpertFNO | $4.70 \times 10^{-3} \pm 1.1 \times 10^{-4}$ | $9.98 \times 10^{-1} \pm 4.1 \times 10^{-2}$ |

equations

$$\frac{\partial \mathbf{u}}{\partial t} + (\mathbf{u} \cdot \nabla)\mathbf{u} = -\nabla p + \nu \Delta \mathbf{u}, \quad \nabla \cdot \mathbf{u} = 0, \quad \mathbf{x} \in \Omega, \ t \in T, \tag{35}$$

$$\mathbf{u} = \mathbf{u}_b, \tag{36}$$

where $\mathbf{u} = (u, v)$ is the velocity field, $p$ is the pressure, and $\nu$ denotes the kinematic viscosity. The Dirichlet boundary condition $\mathbf{u} = \mathbf{u}_b$ enforces the prescribed lid motion.

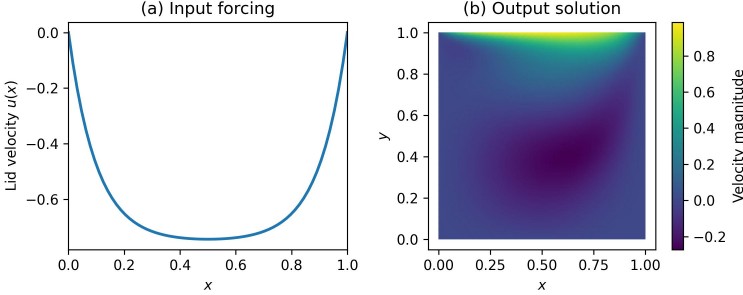

Figure 4: Example input–output pair from the lid-driven cavity flow experiment. (a) One-dimensional lid velocity profile $u(x)$ prescribed along the top boundary of the cavity and used as the model input. (b) Corresponding two-dimensional cavity flow solution for a representative test instance.

We learn an operator mapping from the one-dimensional lid velocity profile prescribed along the top boundary to the resulting steady-state two-dimensional velocity field:

$$\mathcal{G} : u_b(x) \ \mapsto \ \mathbf{u}(x, y).$$

An example input–output pair is shown in Figure 4. We consider the steady-state regime on the unit square domain $\Omega = [0, 1]^2$, following the dataset specification of (Lu et al., 2022). The lid velocity profile is defined as

$$u_b(x) = U\left(1 - \frac{\cosh\left(r(x - \frac{1}{2})\right)}{\cosh\left(\frac{r}{2}\right)}\right), \quad v_b = 0, \tag{37}$$

with $r = 10$. All remaining boundaries are stationary. Steady-state solutions are generated using a lattice Boltzmann method (LBM) solver. The dataset consists of 100 training and 10 test input–output pairs, following (Lu et al., 2022; Sharma & Shankar, 2025).

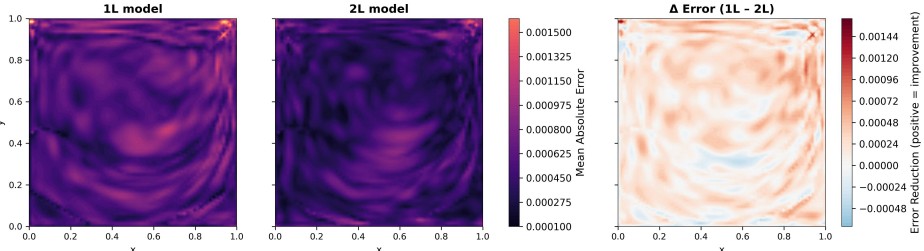

Figure 5: Comparison of single-level (1L) and hierarchical two-level (2L) HiRefPOU DeepONet models for the cavity flow problem. Both models employ six total partitions initialized at identical center locations. The left and center panels show the mean absolute error (MAE) fields for the 1L ($e_{1L}$) and 2L ($e_{2L}$) architectures, respectively. The rightmost panel displays the local error difference $\delta e = e_{1L} - e_{2L}$, where positive values indicate regions in which the hierarchical model improves upon the single-level model.

*Results.* Quantitative results for the lid-driven cavity flow problem are reported in Tables 2 and 3. Among DeepONet-based models, the HiRefPOU variants substantially reduce both relative $\ell_2$ and $\ell_\infty$ errors compared

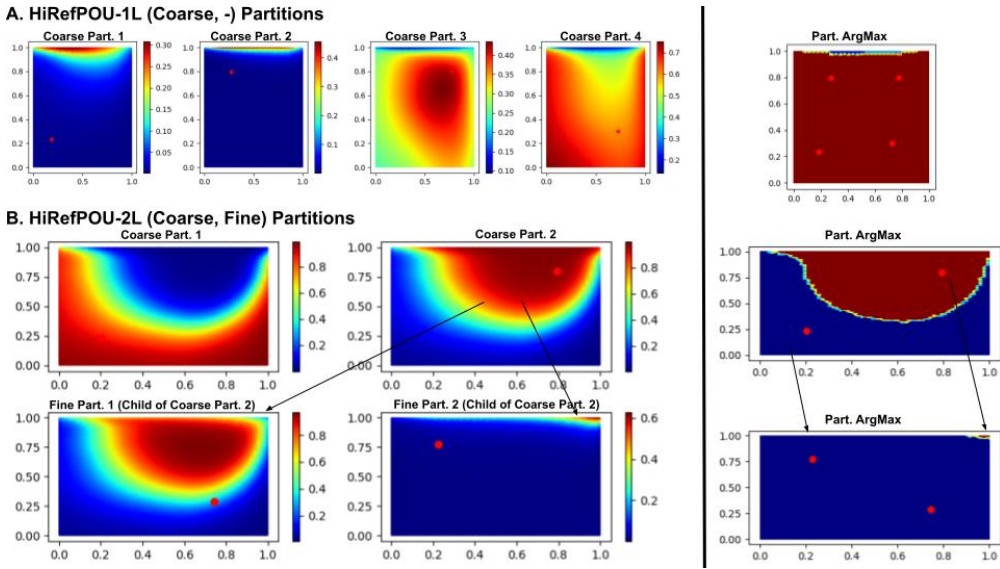

Figure 6: Learned partition-of-unity (POU) gating functions for the cavity flow problem using the single-level (1L) and hierarchical two-level (2L) HiRefPOU-DeepONet models. (A) Single-level (1L) model: spatially varying coarse gating functions associated with individual experts, together with the induced hard partition obtained via the pointwise Argmax. (B) Hierarchical two-level (2L) model: learned coarse-level gating functions (top row) and their corresponding fine-level child partitions (bottom row). Rightmost panels show the induced hard partitions at each level. Colors indicate gating weight magnitude, with warmer colors corresponding to stronger expert activation. The hierarchical model produces localized fine-scale refinements while preserving coherent coarse partitions.

Table 3: Testing $\ell_\infty$ errors, training times, and inference times for the Cavity Flow problem. For each column, the best value is shown in **bold**, and the second-best value is underlined. Ties are bolded when reported values are equal to the displayed precision. Rows corresponding to proposed adaptive PoU-based models are highlighted.

| | Model | $\ell_\infty$ Error | Mean Train Time (s) | Inference Time (s) |
|---|---|---|---|---|
| **DeepONets** | Vanilla | $1.67 \times 10^{-1} \pm 2.7 \times 10^{-2}$ | $\mathbf{1.01 \times 10^{-3} \pm 5.8 \times 10^{-4}}$ | $9.54 \times 10^{-4} \pm 6.0 \times 10^{-4}$ |
| | (P+1)-Vanilla | $7.02 \times 10^{-2} \pm 2.7 \times 10^{-2}$ | $\mathbf{1.01 \times 10^{-3} \pm 7.4 \times 10^{-5}}$ | $\mathbf{1.46 \times 10^{-4} \pm 1.3 \times 10^{-5}}$ |
| | (Vanilla, POU) | $6.33 \times 10^{-2} \pm 2.6 \times 10^{-2}$ | $5.44 \times 10^{-3} \pm 3.8 \times 10^{-3}$ | $5.26 \times 10^{-3} \pm 3.8 \times 10^{-3}$ |
| | (POD, POU) | $4.81 \times 10^{-3} \pm 3.7 \times 10^{-4}$ | $\underline{3.33 \times 10^{-3} \pm 5.3 \times 10^{-5}}$ | $3.20 \times 10^{-3} \pm 7.1 \times 10^{-6}$ |
| | (Vanilla, POD, POU) | $6.95 \times 10^{-3} \pm 2.0 \times 10^{-3}$ | $4.42 \times 10^{-3} \pm 6.6 \times 10^{-5}$ | $4.29 \times 10^{-3} \pm 1.1 \times 10^{-4}$ |
| | HiRefPoU-1L (POD) | $\mathbf{3.03 \times 10^{-3} \pm 5.2 \times 10^{-4}}$ | $3.60 \times 10^{-3} \pm 7.4 \times 10^{-4}$ | $4.74 \times 10^{-4} \pm 1.1 \times 10^{-4}$ |
| | HiRefPoU-2L (POD) | $\underline{3.61 \times 10^{-3} \pm 4.6 \times 10^{-4}}$ | $4.05 \times 10^{-3} \pm 1.1 \times 10^{-3}$ | $4.68 \times 10^{-4} \pm 8.5 \times 10^{-5}$ |
| | HiRefPOU-3L (POD) | $4.85 \times 10^{-3} \pm 5.6 \times 10^{-4}$ | $4.97 \times 10^{-3} \pm 7.7 \times 10^{-7}$ | $\underline{4.57 \times 10^{-4} \pm 1.9 \times 10^{-5}}$ |
| **FNOs** | GeoFNO | $9.63 \times 10^{-2} \pm 2.0 \times 10^{-2}$ | $1.92 \times 10^{-1} \pm 6.3 \times 10^{-2}$ | $1.78 \times 10^{-1} \pm 1.4 \times 10^{-1}$ |
| | LocalFNO | $1.98 \times 10^{-2} \pm 1.1 \times 10^{-3}$ | $4.01 \times 10^{-2} \pm 1.2 \times 10^{-3}$ | $2.37 \times 10^{1} \pm 6.12 \times 10^{-1}$ |
| | PoU-SharedFNO (GeoFNO) | $9.71 \times 10^{-2} \pm 4.1 \times 10^{-2}$ | $5.42 \times 10^{-2} \pm 9.0 \times 10^{-4}$ | $1.88 \times 10^{-2} \pm 2.2 \times 10^{-4}$ |
| | PoU-ExpertFNO (GeoFNO) | $1.24 \times 10^{-2} \pm 8.0 \times 10^{-4}$ | $1.05 \times 10^{-1} \pm 2.8 \times 10^{-5}$ | $2.88 \times 10^{-2} \pm 1.1 \times 10^{-4}$ |

with the vanilla and static-partition baselines. HiRefPOU-1L achieves the lowest relative $\ell_2$ and $\ell_\infty$ errors among the DeepONet-family models, while HiRefPOU-2L gives comparable accuracy. The FNO-based results are mixed: LocalFNO and PoU-ExpertFNO are competitive in relative $\ell_2$, but the HiRefPOU-DeepONet variants achieve the best $\ell_\infty$ errors.

Figure 5 compares the spatial distribution of prediction errors between the two architectures. The hierarchical model reduces error in localized regions of the domain, while maintaining similar global accuracy to the single-level model. Figure 6 further illustrates the learned partition structures, showing that the 1L model learns partitions that align well with dominant flow features, and that the 2L model introduces targeted local refinements within these regions.

Finally, Table 3 reports training and inference times. The adaptive HiRefPOU models incur additional cost relative to the simplest vanilla DeepONet variants, reflecting the overhead of gating and expert evaluation. However, this overhead is moderate within the DeepONet family and accompanies substantial improvements in both relative $\ell_2$ and $\ell_\infty$ error. Thus, the cavity results support an accuracy–cost tradeoff: adaptive partitioning improves predictive accuracy and interpretability, while requiring additional computation relative to the simplest baselines.

### 7.3 2D Reaction-Diffusion Problem

*Problem Setup.* We next consider a two-dimensional reaction–diffusion equation describing the spatiotemporal concentration $c(y,t)$ of a chemical species undergoing binding and unbinding reactions. The governing partial differential equation is

$$\frac{\partial c}{\partial t} = k_{\mathrm{on}}(R - c)\,c_{\mathrm{amb}} - k_{\mathrm{off}}\,c + \nu\Delta c, \qquad y \in \Omega,\ t \in T, \tag{38}$$

with homogeneous Neumann boundary conditions

$$\nu\frac{\partial c}{\partial n} = 0, \qquad y \in \partial\Omega. \tag{39}$$

Here, $k_{\mathrm{on}}$ and $k_{\mathrm{off}}$ denote reaction on- and off-rates, $\nu = 0.1$ is the diffusion coefficient, and $R = 2$ controls the available reactant concentration. The ambient concentration

$$c_{\mathrm{amb}}(y,t) = \big[1 + \cos(2\pi y_1)\cos(2\pi y_2)\big]e^{-\pi t}$$

introduces spatial and temporal variability into the reaction term.

Both reaction coefficients are discontinuous in the horizontal direction $y_1$:

$$k_{\mathrm{on}}(y_1) = \begin{cases} 2, & y_1 \leq 1, \\ 0, & \text{otherwise,} \end{cases} \qquad k_{\mathrm{off}}(y_1) = \begin{cases} 0.2, & y_1 \leq 1, \\ 0, & \text{otherwise.} \end{cases} \tag{40}$$

This discontinuity induces a sharp transition across the interface $y_1 = 1$, producing localized gradients that are challenging for global operator-learning models. Our objective is to learn the operator

$$\mathcal{G} : c(y,0) \mapsto c(y,T), \qquad T = 0.5,$$

mapping the initial concentration field to the solution at the final time. The problem is defined on $\Omega = [0,2]^2$ with initial condition $c(y,0) \sim \mathcal{U}(0,1)$. An example input–output pair is shown in Figure 7. Reference solutions are taken from Sharma & Shankar (2025), computed using an RBF–FD discretization (Shankar & Fogelson, 2018; Shankar et al., 2021). The dataset consists of 2207 spatial collocation points, with 1000 training and 200 test input–output pairs.

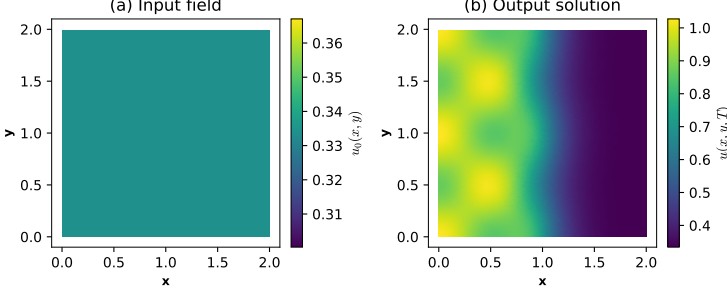

Figure 7: Example input–output pair for the two-dimensional reaction-diffusion problem. (a) Spatial input field $c(y,0)$ provided as input to the operator-learning model. (b) Corresponding solution field $c(y,T)$ at the final time.

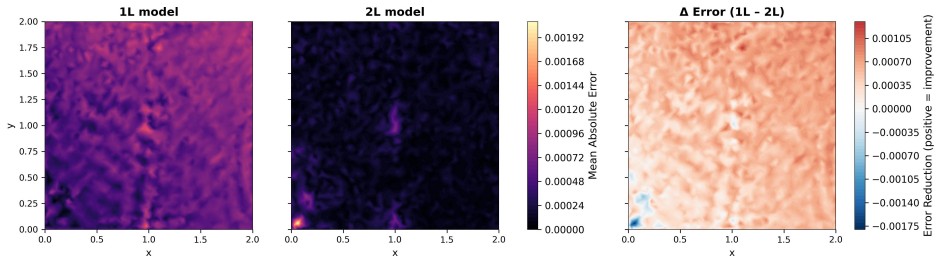

Figure 8: Comparison of single-level (1L) and hierarchical two-level (2L) HiRefPOUDeepONet models. Both models employ six total partitions with identical initial center locations. The left and center panels show the mean absolute error (MAE) fields for the 1L ($e_{1L}$) and 2L ($e_{2L}$) architectures, respectively. The rightmost panel displays the local error difference $\delta e = e_{1L} - e_{2L}$, where positive values indicate regions in which the hierarchical model improves upon the single-level model.

Table 4: Testing $\ell_\infty$ errors, training times, and inference times for the 2D Reaction-Diffusion problem. For each column, the best value is shown in **bold**, and the second-best value is underlined. Rows corresponding to proposed adaptive PoU-based models are highlighted.

| | Model | $\ell_\infty$ Error | Mean Train Time (s) | Inference Time (s) |
|---|---|---|---|---|
| **DeepONets** | Vanilla | $5.65 \times 10^{-3} \pm 1.3 \times 10^{-3}$ | $\mathbf{3.66 \times 10^{-4} \pm 4.3 \times 10^{-6}}$ | $\mathbf{8.74 \times 10^{-5} \pm 4.7 \times 10^{-6}}$ |
| | (P+1)-Vanilla | $1.81 \times 10^{-3} \pm 4.8 \times 10^{-4}$ | $\underline{9.88 \times 10^{-4} \pm 3.2 \times 10^{-5}}$ | $\underline{1.41 \times 10^{-4} \pm 6.1 \times 10^{-6}}$ |
| | (Vanilla, POU) | $2.45 \times 10^{-3} \pm 5.5 \times 10^{-4}$ | $2.84 \times 10^{-3} \pm 8.7 \times 10^{-4}$ | $2.52 \times 10^{-3} \pm 1.2 \times 10^{-3}$ |
| | (POD, POU) | $2.15 \times 10^{-3} \pm 6.4 \times 10^{-4}$ | $6.05 \times 10^{-3} \pm 2.7 \times 10^{-3}$ | $5.75 \times 10^{-3} \pm 3.1 \times 10^{-3}$ |
| | (Vanilla, POD, POU) | $2.46 \times 10^{-3} \pm 2.2 \times 10^{-4}$ | $7.38 \times 10^{-3} \pm 3.5 \times 10^{-4}$ | $7.34 \times 10^{-3} \pm 3.2 \times 10^{-5}$ |
| | HiRefPOU-1L (POD) | $1.22 \times 10^{-3} \pm 3.1 \times 10^{-4}$ | $3.42 \times 10^{-3} \pm 9.6 \times 10^{-4}$ | $3.76 \times 10^{-4} \pm 3.9 \times 10^{-5}$ |
| | HiRefPOU-2L (POD) | $\underline{1.20 \times 10^{-3} \pm 2.5 \times 10^{-4}}$ | $4.72 \times 10^{-3} \pm 1.0 \times 10^{-3}$ | $5.62 \times 10^{-4} \pm 1.3 \times 10^{-4}$ |
| | HiRefPOU-3L (POD) | $\mathbf{1.14 \times 10^{-3} \pm 2.2 \times 10^{-4}}$ | $4.65 \times 10^{-3} \pm 4.0 \times 10^{-5}$ | $5.73 \times 10^{-4} \pm 2.9 \times 10^{-5}$ |
| **FNOs** | GeoFNO | $1.57 \times 10^{-1} \pm 3.3 \times 10^{-1}$ | $5.30 \times 10^{-1} \pm 1.3 \times 10^{-1}$ | $5.44 \times 10^{-2} \pm 5.3 \times 10^{-2}$ |
| | LocalFNO | $2.27 \times 10^{-2} \pm 6.8 \times 10^{-6}$ | $6.94 \times 10^{-2} \pm 1.2 \times 10^{-3}$ | $7.16 \times 10^{-2} \pm 2.7 \times 10^{-3}$ |
| | PoU-SharedFNO (GeoFNO) | $2.50 \times 10^{-2} \pm 3.5 \times 10^{-3}$ | $5.35 \times 10^{-2} \pm 9.4 \times 10^{-5}$ | $1.89 \times 10^{-2} \pm 9.2 \times 10^{-5}$ |
| | PoU-ExpertFNO (GeoFNO) | $7.00 \times 10^{-2} \pm 1.5 \times 10^{-3}$ | $7.51 \times 10^{-1} \pm 1.2 \times 10^{-4}$ | $4.19 \times 10^{-2} \pm 1.1 \times 10^{-5}$ |

*Results.* We next evaluate the effect of hierarchical partitioning by comparing the single-level (1L), two-level (2L), and three-level (3L) HiRefPOU DeepONet models on the 2D reaction–diffusion benchmark. The 1L and 2L models use the same total number of partitions and comparable parameter counts, so their performance differences isolate the effect of organizing the same partition budget hierarchically. The 3L model uses the same total number of leaf partitions, allowing us to test whether deeper hierarchical refinement provides additional benefit without simply increasing the finest-level partition count. Quantitative results are reported in Tables 2 and 4.

The adaptive HiRefPOU-DeepONet models improve over the vanilla and static-partition DeepONet baselines in both relative $\ell_2$ and $\ell_\infty$ error. In the relative $\ell_2$ results, HiRefPOU-3L achieves the lowest error, followed by HiRefPOU-2L. The same trend appears in the $\ell_\infty$ results, where the 3L and 2L variants provide the best DeepONet-family performance. These results suggest that hierarchical refinement is beneficial for this problem, where discontinuous reaction coefficients create localized solution features near the interface $y_1 = 1$. However, the gains from 3L over 2L are modest, and the timing results show that additional hierarchy increases computational cost relative to the simplest DeepONet models. Thus, the benefit of hierarchy should be interpreted as an accuracy–cost tradeoff rather than a uniformly free improvement.

Figure 8 visualizes the spatial distribution of prediction errors for the 1L and 2L models. The hierarchical model reduces error near the interface at $y_1 = 1$, where sharp gradients arise due to coefficient discontinuities. Figure 9 shows the corresponding learned POU gating functions, illustrating that the 2L model introduces targeted fine-scale refinements in these regions while preserving coherent coarse partitions elsewhere. Together with the quantitative results, these visualizations indicate that hierarchical partitioning is most useful when the learned refinements align with localized structures in the solution operator.

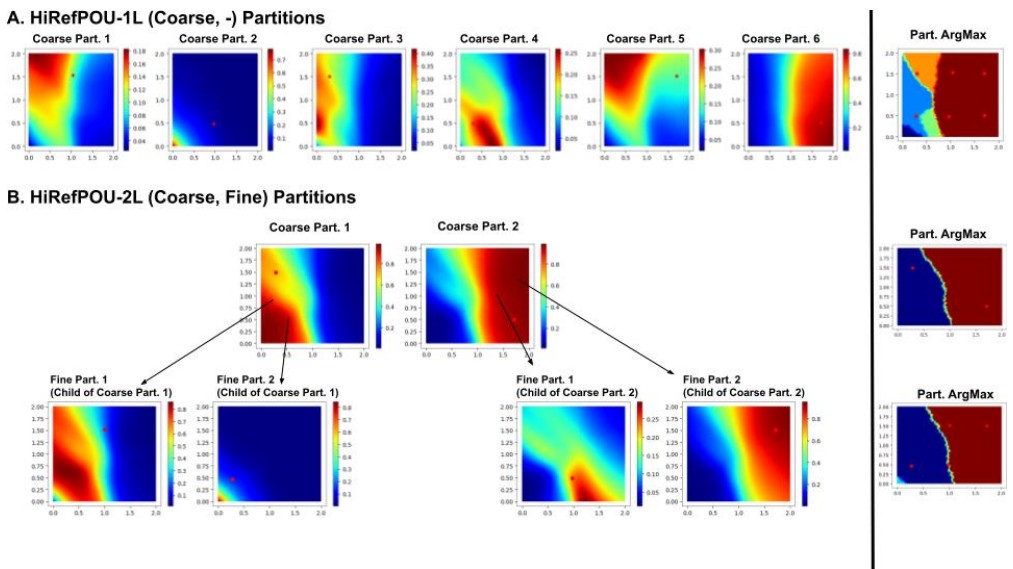

Figure 9: Comparison of single-level (1L) and hierarchical two-level (2L) HiRefPOU DeepONet learned partitions. Each panel shows the learned POU gating functions corresponding to different experts within the network. The top row illustrates the partitions produced by the single-level (1L) model, while the middle and bottom rows correspond to the hierarchical two-level (2L) model, where coarse partitions are further subdivided into locally refined subregions. The hierarchical model introduces localized refinements near the coefficient discontinuity while preserving coherent coarse partitions across the domain.

### 7.4 3D Reaction-Diffusion with Variable Diffusion Coefficient

*Problem Setup.* We next consider a three-dimensional reaction-diffusion system with a spatially varying diffusion coefficient, posed on a curved domain. Let $c(y, t)$ denote the concentration field with $y \in \mathbb{R}^3$. The governing equation is

$$\frac{\partial c}{\partial t} = k_{\text{on}}(R - c)c_{\text{amb}} - k_{\text{off}}c + \nabla \cdot \big(K(y)\nabla c\big), \qquad y \in \Omega, \ t \in T, \tag{41}$$

with homogeneous Neumann boundary conditions

$$K(y)\frac{\partial c}{\partial n} = 0, \qquad y \in \partial\Omega. \tag{42}$$

The computational domain is the unit ball

$$\Omega = \{y \in \mathbb{R}^3 : \|y\|_2 \le 1\},$$

with boundary $\partial\Omega = \mathbb{S}^2$, and the time interval is $T = [0, 0.5]$. An example input–output pair is shown in Figure 10. The reaction coefficients $k_{\text{on}}$ and $k_{\text{off}}$ follow the same configuration as in the two-dimensional case, taking nonzero values in the half-space $y_1 \le 0$ and vanishing elsewhere. The ambient source term is defined as

$$c_{\text{amb}}(y, t) = \big[1 + \cos(2\pi y_1)\cos(2\pi y_2)\sin(2\pi y_3)\big]e^{-\pi t}.$$

The spatially varying diffusion coefficient $K(y)$ is designed to introduce sharp gradients,

$$K(y_1) = B + \frac{C}{\tanh(A)}\Big[(A - 3)\tanh(8y_1 - 5) - (A - 15)\tanh(8y_1 + 5) + A\tanh(A)\Big], \tag{43}$$

with $A = 9$, $B = 0.0215$, and $C = 0.005$. This configuration yields strong spatial heterogeneity in the diffusion operator. Our objective is to learn the solution operator

$$\mathcal{G} : c(y, 0) \mapsto c(y, 0.5).$$

Reference solutions are taken from Sharma & Shankar (2025) and computed using a high-order RBF-FD solver (Shankar & Fogelson, 2018; Shankar et al., 2021). The dataset consists of 1000 training and 200 test samples, with each function evaluated at 4325 spatial collocation points.

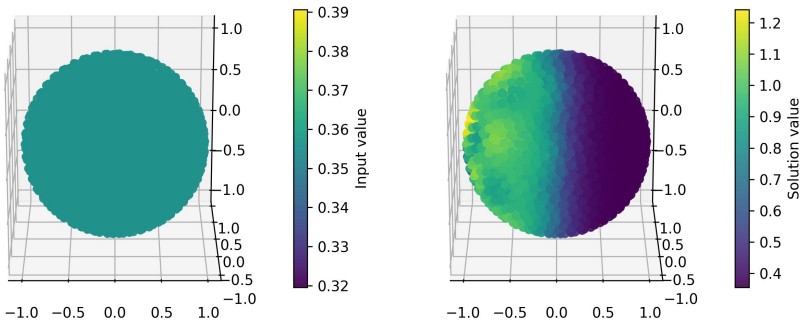

Figure 10: Example input–output pair for the three-dimensional reaction-diffusion problem. (Left) Spatial input field provided as input to the operator-learning model. (Right) Corresponding solution field at the final time.

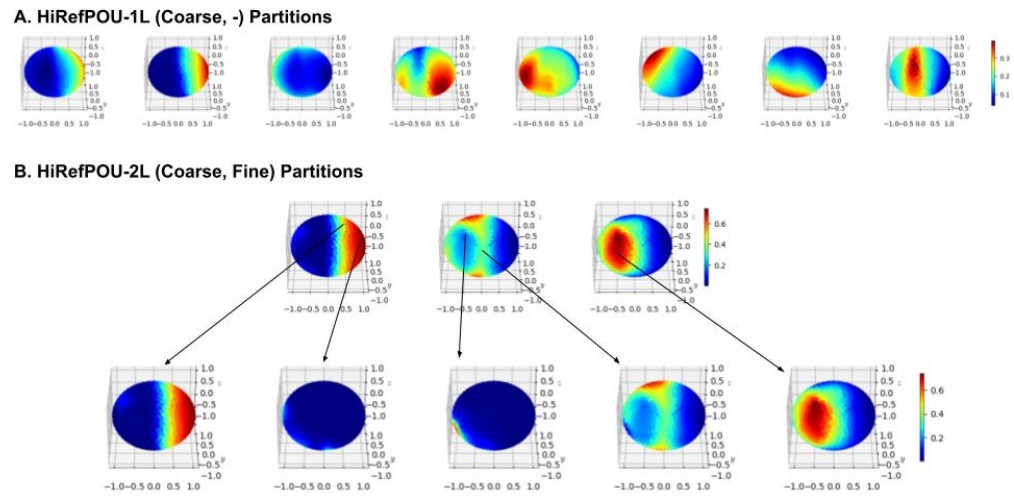

Figure 11: Comparison of single-level (1L) and hierarchical two-level (2L) HiRefPOU DeepONet learned partitions. Each panel shows the learned POU gating functions corresponding to different experts within the network. The top row illustrates the partitions produced by the single-level (1L) model, while the middle and bottom rows correspond to the hierarchical two-level (2L) model, where coarse partitions are further subdivided into locally refined subregions. The hierarchical model introduces localized refinements aligned with regions of strong diffusion variation while preserving coherent coarse partitions across the ball domain.

*Results.* Quantitative results for the three-dimensional reaction–diffusion problem are reported in Tables 2 and 5. Among DeepONet-based models, the HiRefPOU variants outperform the vanilla and static-partition baselines, demonstrating the effectiveness of adaptive PoU representations for volumetric heterogeneous problems. HiRefPOU-2L achieves the lowest $\ell_\infty$ error, while HiRefPOU-3L achieves the lowest relative $\ell_2$ error in the main comparison table. This indicates that hierarchical refinement improves accuracy, although the optimal number of levels depends on the metric.

Figure 11 visualizes the learned POU gating functions. The single-level model produces smooth global partitions over the ball, while the hierarchical model introduces localized refinements aligned with regions of strong diffusion variation. The FNO-based models perform substantially worse on this benchmark,

Table 5: Testing $\ell_\infty$ errors, training times, and inference times for the 3D Reaction-Diffusion problem. For each column, the best value is shown in **bold**, and the second-best value is underlined. Rows corresponding to proposed adaptive PoU-based models are highlighted.

| | Model | $\ell_\infty$ Error | Mean Train Time (s) | Inference Time (s) |
|---|---|---|---|---|
| **DeepONets** | Vanilla | $3.81 \times 10^{-3} \pm 1.5 \times 10^{-3}$ | $\mathbf{4.61 \times 10^{-4} \pm 1.0 \times 10^{-4}}$ | $4.50 \times 10^{-4} \pm 1.2 \times 10^{-4}$ |
| | (P+1)-Vanilla | $9.77 \times 10^{-2} \pm 1.1 \times 10^{-1}$ | $\underline{1.69 \times 10^{-3} \pm 5.7 \times 10^{-4}}$ | $1.64 \times 10^{-3} \pm 5.9 \times 10^{-4}$ |
| | (Vanilla, PoU) | $1.13 \times 10^{-3} \pm 5.1 \times 10^{-4}$ | $1.40 \times 10^{-2} \pm 6.7 \times 10^{-3}$ | $1.39 \times 10^{-2} \pm 6.7 \times 10^{-3}$ |
| | (POD, PoU) | $1.83 \times 10^{-3} \pm 5.2 \times 10^{-4}$ | $1.11 \times 10^{-2} \pm 9.7 \times 10^{-5}$ | $1.09 \times 10^{-2} \pm 4.8 \times 10^{-5}$ |
| | (Vanilla, POD, PoU) | $1.36 \times 10^{-3} \pm 5.0 \times 10^{-4}$ | $1.13 \times 10^{-2} \pm 1.9 \times 10^{-4}$ | $1.11 \times 10^{-2} \pm 1.1 \times 10^{-4}$ |
| | HiRefPOU-1L (POD) | $6.87 \times 10^{-4} \pm 4.1 \times 10^{-4}$ | $2.63 \times 10^{-3} \pm 1.0 \times 10^{-4}$ | $\mathbf{3.54 \times 10^{-4} \pm 2.7 \times 10^{-5}}$ |
| | HiRefPOU-2L (POD) | $\mathbf{4.75 \times 10^{-4} \pm 1.9 \times 10^{-4}}$ | $2.86 \times 10^{-3} \pm 1.3 \times 10^{-4}$ | $\underline{4.38 \times 10^{-4} \pm 3.3 \times 10^{-5}}$ |
| | HiRefPOU-3L (POD) | $\underline{4.95 \times 10^{-4} \pm 5.2 \times 10^{-5}}$ | $7.13 \times 10^{-3} \pm 5.7 \times 10^{-5}$ | $6.16 \times 10^{-4} \pm 8.7 \times 10^{-6}$ |
| **FNOs** | GeoFNO | $1.80 \pm 7.3 \times 10^{-1}$ | $5.61 \pm 8.9 \times 10^{-3}$ | $2.52 \times 10^{-1} \pm 2.5 \times 10^{-2}$ |
| | LocalFNO | $3.20 \times 10^{-2} \pm 1.1 \times 10^{-3}$ | $1.51 \times 10^{-1} \pm 8.4 \times 10^{-3}$ | $8.36 \times 10^{-2} \pm 2.5 \times 10^{-3}$ |
| | PoU-SharedFNO (GeoFNO) | $1.09 \pm 5.6 \times 10^{-2}$ | $7.35 \pm 9.4 \times 10^{-2}$ | $4.89 \times 10^{-1} \pm 9.2 \times 10^{-2}$ |
| | PoU-ExpertFNO (GeoFNO) | $7.09 \times 10^{-1} \pm 1.6 \times 10^{-2}$ | $1.05 \times 10^1 \pm 5.2 \times 10^{-1}$ | $1.09 \pm 1.7 \times 10^{-1}$ |

suggesting that the combination of volumetric geometry, spatially varying diffusion, and localized gradients is less naturally matched to the tested Fourier backbones. Overall, these results highlight the advantage of hierarchical localized operator-learning architectures for heterogeneous three-dimensional reaction–diffusion systems.

### 7.5 1D Burgers' Equation

We next consider two one-dimensional Burgers benchmarks. The first is the standard periodic Gaussian-random-field benchmark used widely in operator-learning studies. The second is a modified multiscale periodic benchmark, constructed to contain more explicit mixed coarse- and fine-scale spatial structure. The purpose of the second benchmark is to more directly probe the regimes in which hierarchical adaptive partitioning is beneficial.

*Problem Setup.* We next consider the one-dimensional Burgers' equation:

$$\frac{\partial u}{\partial t} + u\frac{\partial u}{\partial x} = \nu\frac{\partial^2 u}{\partial x^2}, \quad x \in (0,1), \ t \in (0,1] \tag{44}$$

with periodic boundary conditions and viscosity $\nu = 0.001$. Examples of final-time solutions are shown in Figure 13.

In this problem, we learn the operator mapping from the initial condition $u(x,0) = u_0(x)$ to the solution $u(x,t)$ at $t = 1$,

$$\mathcal{G} : u_0(x) \mapsto u(x,1). \tag{45}$$

Similar to (Lu et al., 2022), we generate each dataset such that each initial condition is generated according to a random distribution:

$$\mu = \mathcal{N}(0, 625(-\Delta + 25I)^{-2}). \tag{46}$$

The initial condition $u_0$ is modeled as a sample from a Gaussian random field with mean zero and covariance operator

$$C = \sigma^2 \left(-\Delta + \tau^2 I\right)^{-\gamma}, \tag{47}$$

where $\sigma^2 = 25$, $\tau = 5$, and $\gamma = 2.5$. The field is defined on the periodic domain $(0,1)$, and realizations are generated spectrally by drawing independent standard normal coefficients $\xi_k, \eta_k \sim \mathcal{N}(0,1)$ and scaling by the eigenvalues of the covariance operator:

$$\lambda_k = \sqrt{2}|\sigma| \left((2\pi k)^2 + \tau^2\right)^{-\gamma/2}.$$

The resulting Fourier coefficients are combined using the `Chebfun` package in MATLAB to construct smooth periodic realizations of the form

$$u_0(x) = \sum_{k=1}^{K} \alpha_k \cos(2\pi k x) + \beta_k \sin(2\pi k x),$$

where $K$ denotes the number of retained Fourier modes and

$$\alpha_k = \lambda_k \xi_k, \qquad \beta_k = \lambda_k \eta_k, \qquad \xi_k, \eta_k \overset{\text{i.i.d.}}{\sim} \mathcal{N}(0,1).$$

To generate the dataset, each realization of $u_0$ is evolved forward in time according to the Burgers' equation using a spectral solver implemented in `Chebfun`. This method employs adaptive Fourier expansions in space and high-order time-stepping schemes to ensure spectral accuracy. We discretize the spatial domain uniformly with $2^{13} = 8192$ collocation points and evolve the system over 500 uniform time steps up to $t = 1$. For each realization, we record both the initial condition $u_0(x)$ and the final-time solution $u(x, 1)$. This is a common benchmark problem for operator learning methods (Lu et al., 2022).

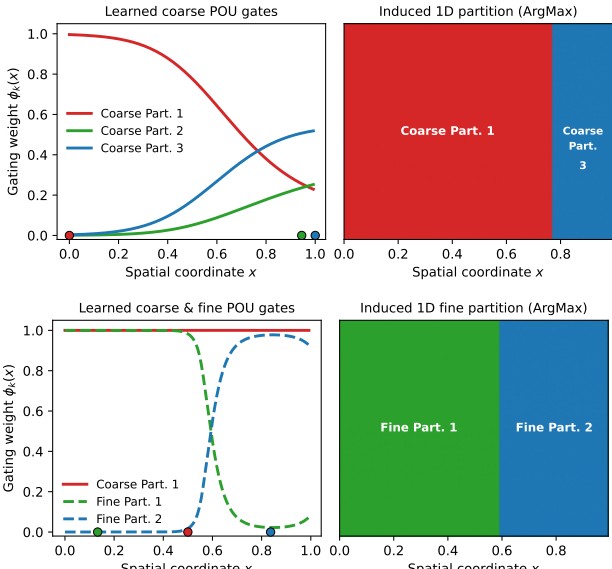

Figure 12: *(1D Burgers Problem.)* Learned partition-of-unity (POU) gates for HiRefPOU-DeepONet models. (Top left) Continuous coarse-level gating functions $\{\phi_j(x)\}_{j=1}^{3}$ learned by the HiRefPOU-1L model. Colored markers indicate the learned center locations associated with each partition. (Top right) Induced hard partition obtained via $\arg\max_j \phi_j(x)$, illustrating the discrete domain decomposition implied by the learned soft POU representation. (Bottom left) Continuous coarse- and fine-level gating functions $\{\phi_j^c(x)\}_{j=1}^{1}$ and $\{\phi_k^f(x)\}_{k=1}^{2}$ learned by the HiRefPOU-2L model. (Bottom right) Induced hard fine-level partition obtained via $\arg\max_k \phi_k^f(x)$.

*Results.* Table 2 reports testing relative $\ell_2$ errors, and Table 6 reports testing $\ell_\infty$ errors, training times, and inference times for all evaluated operator-learning models on the 1D Burgers benchmark. Among DeepONet-based models, the primary improvement comes from introducing adaptive spatial localization through HiRefPOU-1L. The 2L and 3L variants remain competitive, but they provide only modest additional improvement on this dataset. This suggests that, for the standard Burgers benchmark, much of the relevant operator structure at the final time is already captured by a single adaptive partition level. In other words, the benchmark benefits from localization, but does not appear to require a deeper nested hierarchy of spatial refinements.

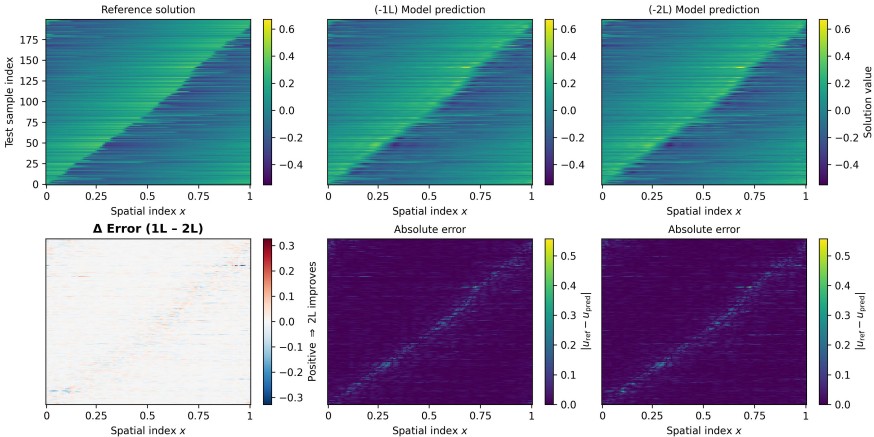

Figure 13: *(1D Burgers Problem.)* Operator learning predictions at final time $t = 1$. (Top row) Reference solution $u(x, 1)$, HiRefPOU-1L prediction, and HiRefPOU-2L prediction, evaluated over the full test set. Each heatmap displays solution values as a function of spatial coordinate $x \in [0, 1]$ (horizontal axis) and test sample index (vertical axis). (Bottom row) Point-wise absolute error fields $|u_{\mathrm{ref}} - u_{\mathrm{pred}}|$ for the 1L and 2L models, along with their difference. Positive values indicate locations where the 2L model achieves lower error than the 1L model.

Table 6: Testing $\ell_\infty$ errors, training times, and inference times for the 1D Burgers problem. For each column, the best value is shown in **bold**, and the second-best value is underlined. For PoU-FNO variants, the FNO backbone used by the model is indicated in parentheses. Rows corresponding to proposed adaptive PoU-based models are highlighted.

| | Model | $\ell_\infty$ Error | Mean Train Time (s) | Inference Time (s) |
|---|---|---|---|---|
| **DeepONets** | Vanilla | $5.17 \times 10^{-1} \pm 4.9 \times 10^{-3}$ | $\mathbf{2.61 \times 10^{-4} \pm 5.6 \times 10^{-6}}$ | $\mathbf{5.56 \times 10^{-5} \pm 5.8 \times 10^{-7}}$ |
| | (P+1)-Vanilla | $5.35 \times 10^{-1} \pm 1.1 \times 10^{-2}$ | $5.59 \times 10^{-4} \pm 1.7 \times 10^{-5}$ | $1.30 \times 10^{-4} \pm 2.1 \times 10^{-5}$ |
| | (Vanilla, PoU) | $5.65 \times 10^{-1} \pm 2.1 \times 10^{-2}$ | $6.60 \times 10^{-4} \pm 2.5 \times 10^{-5}$ | $9.46 \times 10^{-5} \pm 6.5 \times 10^{-6}$ |
| | (POD, PoU) | $4.79 \times 10^{-1} \pm 5.2 \times 10^{-3}$ | $\underline{5.36 \times 10^{-4} \pm 1.8 \times 10^{-5}}$ | $\underline{9.22 \times 10^{-5} \pm 2.6 \times 10^{-6}}$ |
| | (Vanilla, POD, PoU) | $4.83 \times 10^{-1} \pm 6.7 \times 10^{-3}$ | $6.60 \times 10^{-4} \pm 2.2 \times 10^{-5}$ | $9.27 \times 10^{-5} \pm 2.8 \times 10^{-6}$ |
| | HiRefPoU-1L (POD) | $4.57 \times 10^{-1} \pm 4.6 \times 10^{-3}$ | $1.36 \times 10^{-3} \pm 7.8 \times 10^{-5}$ | $1.55 \times 10^{-4} \pm 8.0 \times 10^{-6}$ |
| | HiRefPoU-2L (POD) | $4.56 \times 10^{-1} \pm 5.3 \times 10^{-3}$ | $2.46 \times 10^{-3} \pm 5.4 \times 10^{-4}$ | $2.29 \times 10^{-4} \pm 3.4 \times 10^{-5}$ |
| | HiRefPOU-3L (POD) | $4.56 \times 10^{-1} \pm 7.9 \times 10^{-3}$ | $1.62 \times 10^{-3} \pm 8.6 \times 10^{-5}$ | $2.66 \times 10^{-4} \pm 1.3 \times 10^{-5}$ |
| **FNOs** | GeoFNO | $7.98 \times 10^{-1} \pm 4.2 \times 10^{-2}$ | $2.39 \times 10^{-2} \pm 2.7 \times 10^{-2}$ | $2.22 \times 10^{-3} \pm 7.2 \times 10^{-4}$ |
| | LocalFNO | $\mathbf{2.77 \times 10^{-1} \pm 1.0 \times 10^{-1}}$ | $8.07 \times 10^{-2} \pm 2.5 \times 10^{-3}$ | $7.30 \times 10^{-2} \pm 4.3 \times 10^{-4}$ |
| | PoU-SharedFNO (GeoFNO) | $6.19 \times 10^{-1} \pm 1.7 \times 10^{-1}$ | $4.19 \times 10^{-3} \pm 1.5 \times 10^{-4}$ | $2.07 \times 10^{-3} \pm 1.3 \times 10^{-5}$ |
| | PoU-ExpertFNO (GeoFNO) | $4.30 \times 10^{-1} \pm 9.3 \times 10^{-2}$ | $1.09 \times 10^{-2} \pm 3.6 \times 10^{-5}$ | $4.78 \times 10^{-3} \pm 2.2 \times 10^{-5}$ |
| | PoU-SharedFNO (LocalFNO) | $4.01 \times 10^{-1} \pm 8.9 \times 10^{-2}$ | $5.96 \times 10^{-2} \pm 1.1 \times 10^{-3}$ | $3.70 \times 10^{-3} \pm 1.1 \times 10^{-4}$ |
| | PoU-ExpertFNO (LocalFNO) | $\underline{3.07 \times 10^{-1} \pm 3.2 \times 10^{-2}}$ | $1.01 \times 10^{-1} \pm 6.8 \times 10^{-3}$ | $5.38 \times 10^{-3} \pm 9.8 \times 10^{-5}$ |

Appendix Tables 9 and 10 further clarify the behavior of adaptive partitioning on the standard Burgers benchmark. For the 1L model, the best results are obtained using the hybrid gate formulation. The partition-count sensitivity study shows that a small number of experts is sufficient, with $M = 2$ achieving the best overall 1L performance among the tested settings. Taken together, these ablations support the conclusion that the primary gain on the standard Burgers benchmark comes from introducing adaptive localization itself, while increasing the number of partitions or adding deeper hierarchical refinement provides only modest additional benefit.

The strongest overall results, however, are obtained by LocalFNO. This behavior is consistent with the structure of the standard Burgers benchmark. The problem is posed on a periodic one-dimensional domain with data evaluated on a regular grid, and the input and output fields are generated from smooth periodic random fields evolved by a low-viscosity advection–diffusion equation. These properties are well matched to Fourier-based convolutional operator models: the periodic boundary conditions align naturally with spectral representations, while the regular grid allows convolutional and Fourier layers to exploit translation structure efficiently. Moreover, the final-time Burgers solutions contain sharp gradients and phase-shifted advective

features, which can be difficult for pointwise DeepONet-style branch–trunk factorizations to represent unless substantial local capacity is introduced. LocalFNO directly builds in a local spectral inductive bias, making it especially effective for this benchmark.

The PoU-FNO variants remain competitive, but they do not outperform LocalFNO on this problem. This suggests that, for the standard 1D Burgers benchmark, the LocalFNO backbone already provides much of the useful locality and spectral structure needed to approximate the solution operator. Adding an additional PoU decomposition can improve flexibility, but it may also introduce extra gating and expert complexity that is not necessary when the dominant structure is already well captured by local Fourier filtering. Thus, the Burgers results should not be interpreted as a failure of adaptive partitioning; rather, they show that the benefit of PoU localization is problem- and backbone-dependent. For regular periodic advection-dominated problems, a LocalFNO backbone can be the more natural inductive bias, whereas the HiRefPOU-DeepONet variants are most beneficial when the operator contains localized spatial heterogeneity that is less aligned with global or convolutional Fourier representations.

### 7.6 Full-Field Burgers Benchmark

To more directly probe the value of deeper hierarchical partitioning, we introduce a synthetic multiscale Burgers benchmark on the periodic domain $\Omega = [-1, 1]$. Unlike the standard Burgers benchmark, which maps an initial condition to a single final-time solution profile, here the learning task is to map a one-dimensional initial condition to the full solution field over space-time. The dataset is constructed so that each realization contains nested coarse-, medium-, and fine-scale localized structure, thereby providing a setting in which additional hierarchy in the model can be meaningfully tested.

*Problem Setup.* We consider the viscous Burgers equation

$$\frac{\partial u}{\partial t} + u\frac{\partial u}{\partial x} = \nu\frac{\partial^2 u}{\partial x^2}, \qquad x \in [-1, 1], \quad t \in [0, 1], \tag{48}$$

with periodic boundary conditions and viscosity $\nu = 10^{-3}$. For each sample, the operator-learning objective is to map the initial condition $u_0(x)$ to the full solution field $u(x,t)$.

Each initial condition is generated as the superposition of a smooth low-frequency periodic function and a nested three-level hierarchy of localized oscillatory packets,

$$u_0(x) = u_{\text{bg}}(x) + \sum_{\ell=1}^{3} b_\ell B(x; c_\ell, w_\ell) \sin(2\pi k_\ell x + \phi_\ell), \tag{49}$$

where the background term is given by

$$u_{\text{bg}}(x) = a_1 \sin(2\pi x + p_1) + a_2 \cos(4\pi x + p_2) + a_3 \sin(6\pi x + p_3), \tag{50}$$

and the periodic envelope function is

$$B(x; c, w) = \exp\left(-\frac{\sin^2(\pi(x - c))}{w^2}\right). \tag{51}$$

Here $a_1, a_2, a_3, b_1, b_2, b_3$, and the phases $p_1, p_2, p_3, \phi_1, \phi_2, \phi_3$ are randomly sampled. The packet centers are chosen hierarchically, with $c_2$ sampled near $c_1$ and $c_3$ sampled near $c_2$, so that the medium-scale feature is spatially nested inside the broad-scale feature and the fine-scale feature is nested inside the medium-scale feature. The widths and frequencies satisfy

$$w_1 > w_2 > w_3, \qquad k_1 < k_2 < k_3, \tag{52}$$

so that the three localized packets represent broad, medium, and fine oscillatory structure, respectively.

This construction yields a family of periodic initial conditions with nested multiscale organization. The broad packet determines a coarse active region, the medium packet refines that region, and the fine packet

Table 7: Testing $\ell_\infty$ errors, training times, and inference times for the Full-Field Burgers problem. For each column, the best value is shown in **bold**, and the second-best value is underlined. For PoU-FNO variants, the FNO backbone used by the model is indicated in parentheses. Rows corresponding to proposed adaptive PoU-based models are highlighted.

| | Model | Full-Field $\ell_\infty$ Error | Mean Train Time (s) | Inference Time (s) |
|---|---|---|---|---|
| **DeepONets** | Vanilla | $3.02 \times 10^{-1} \pm 5.7 \times 10^{-3}$ | $\mathbf{2.79 \times 10^{-4} \pm 7.9 \times 10^{-6}}$ | $\mathbf{1.31 \times 10^{-4} \pm 1.1 \times 10^{-4}}$ |
| | (P+1)-Vanilla | $2.90 \times 10^{-1} \pm 5.0 \times 10^{-3}$ | $\underline{1.08 \times 10^{-3} \pm 2.6 \times 10^{-5}}$ | $9.11 \times 10^{-4} \pm 3.8 \times 10^{-4}$ |
| | (Vanilla, POU) | $2.72 \times 10^{-1} \pm 3.4 \times 10^{-3}$ | $1.47 \times 10^{-2} \pm 1.6 \times 10^{-5}$ | $1.47 \times 10^{-2} \pm 1.6 \times 10^{-5}$ |
| | (POD, POU) | $2.69 \times 10^{-1} \pm 4.2 \times 10^{-3}$ | $1.45 \times 10^{-2} \pm 7.4 \times 10^{-5}$ | $1.46 \times 10^{-2} \pm 3.2 \times 10^{-5}$ |
| | (Vanilla, POD, POU) | $2.84 \times 10^{-1} \pm 4.2 \times 10^{-3}$ | $4.87 \times 10^{-2} \pm 4.2 \times 10^{-3}$ | $4.90 \times 10^{-2} \pm 1.2 \times 10^{-3}$ |
| | HiRefPOU-1L (POD) | $2.72 \times 10^{-1} \pm 6.7 \times 10^{-3}$ | $1.50 \times 10^{-3} \pm 2.6 \times 10^{-5}$ | $\underline{3.57 \times 10^{-4} \pm 6.0 \times 10^{-5}}$ |
| | HiRefPOU-2L (POD) | $2.69 \times 10^{-1} \pm 5.1 \times 10^{-3}$ | $3.59 \times 10^{-3} \pm 6.0 \times 10^{-4}$ | $5.50 \times 10^{-4} \pm 7.1 \times 10^{-5}$ |
| | HiRefPOU-3L (POD) | $2.66 \times 10^{-1} \pm 5.5 \times 10^{-3}$ | $1.49 \times 10^{-2} \pm 1.6 \times 10^{-3}$ | $2.12 \times 10^{-3} \pm 3.6 \times 10^{-4}$ |
| **FNOs** | GeoFNO | $8.26 \times 10^{-2} \pm 8.0 \times 10^{-3}$ | $3.97 \pm 1.6 \times 10^{-3}$ | $4.36 \times 10^{-1} \pm 1.6 \times 10^{-4}$ |
| | LocalFNO | $6.47 \times 10^{-2} \pm 7.4 \times 10^{-3}$ | $1.46 \times 10^{-1} \pm 1.3 \times 10^{-4}$ | $7.33 \times 10^{-2} \pm 4.7 \times 10^{-4}$ |
| | PoU-SharedFNO (LocalFNO) | $\mathbf{5.28 \times 10^{-2} \pm 5.0 \times 10^{-3}}$ | $1.59 \times 10^{-1} \pm 1.6 \times 10^{-3}$ | $7.59 \times 10^{-2} \pm 1.1 \times 10^{-4}$ |
| | PoU-ExpertFNO (LocalFNO) | $\underline{5.82 \times 10^{-2} \pm 6.8 \times 10^{-3}}$ | $2.38 \times 10^{-1} \pm 1.6 \times 10^{-3}$ | $7.94 \times 10^{-2} \pm 1.2 \times 10^{-4}$ |

introduces highly localized oscillatory detail inside it. Because the locations, widths, amplitudes, and phases are randomized from sample to sample, the dataset does not reduce to a fixed template, while still consistently exhibiting hierarchical localized structure. We use this benchmark to assess whether deeper adaptive partitioning provides measurable benefits over shallower adaptive models and non-adaptive models.

*Results.* Table 2 reports the relative $\ell_2$ errors, while Table 7 reports full-field $\ell_\infty$ errors and timing results. Within the DeepONet family, the HiRefPOU variants improve over the vanilla and static-partition baselines, indicating that adaptive localization remains beneficial when learning the full space–time solution field. However, the strongest overall results on this benchmark are obtained by the PoU-FNO variants with a LocalFNO backbone. This suggests that, for periodic grid-aligned space–time problems, local Fourier representations provide a particularly strong inductive bias, and the PoU mechanism is most effective when combined with an appropriate Fourier backbone.

### 7.7 Hierarchical Multiscale Kuramoto–Sivashinsky Benchmark

To complement the Burgers experiments with a more dynamically unstable benchmark, we consider the one-dimensional Kuramoto–Sivashinsky (KS) equation on a periodic domain. The KS equation exhibits mixed-scale spatiotemporal behavior, including instability growth, nonlinear interaction, and localized fine-scale structure, making it a natural testbed for hierarchical adaptive operator learning. In contrast to the standard Burgers benchmark, where the output is often a single final-time solution profile, here the learning task is to map an initial condition to the full solution field over space-time.

*Problem Setup.* We study the periodic KS equation

$$\frac{\partial u}{\partial t} + u\frac{\partial u}{\partial x} + \frac{\partial^2 u}{\partial x^2} + \frac{\partial^4 u}{\partial x^4} = 0, \qquad x \in [0, L], \quad t \in [0, T], \tag{53}$$

with periodic boundary conditions. For each sample, the operator-learning task is to map a one-dimensional initial condition $u_0(x)$ to the full solution field $u(x, t)$.

We construct the initial conditions from a randomized three-level hierarchy of localized oscillatory structure superimposed on a smooth low-frequency periodic background. Specifically, each initial condition is of the form

$$u_0(x) = u_{\text{bg}}(x) + \sum_{\ell=1}^{3} b_\ell B(\xi; c_\ell, w_\ell) \sin(2\pi k_\ell \xi + \phi_\ell), \qquad \xi = x/L, \tag{54}$$

where the background component is given by

$$u_{\text{bg}}(x) = a_1 \sin(2\pi\xi + p_1) + a_2 \cos(4\pi\xi + p_2) + a_3 \sin(6\pi\xi + p_3), \tag{55}$$

Table 8: Testing $\ell_\infty$ errors, training times, and inference times for the KS problem. For each column, the best value is shown in **bold**, and the second-best value is underlined. For PoU-FNO variants, the FNO backbone used by the model is indicated in parentheses. Rows corresponding to proposed adaptive PoU-based models are highlighted.

| | Model | Full-Field $\ell_\infty$ Error | Mean Train Time (s) | Inference Time (s) |
|---|---|---|---|---|
| **DeepONets** | (Vanilla) | $2.45 \pm 3.2 \times 10^{-2}$ | $\mathbf{7.49 \times 10^{-4} \pm 7.8 \times 10^{-5}}$ | $7.58 \times 10^{-4} \pm 2.3 \times 10^{-4}$ |
| | (P+1)-Vanilla | $2.09 \pm 3.2 \times 10^{-2}$ | $2.22 \times 10^{-3} \pm 6.6 \times 10^{-5}$ | $2.26 \times 10^{-3} \pm 2.3 \times 10^{-5}$ |
| | (Vanilla, PoU) | $1.96 \pm 3.5 \times 10^{-2}$ | $2.09 \times 10^{-2} \pm 2.9 \times 10^{-4}$ | $2.10 \times 10^{-2} \pm 1.7 \times 10^{-4}$ |
| | (POD, PoU) | $1.87 \pm 2.5 \times 10^{-2}$ | $1.31 \times 10^{-2} \pm 4.6 \times 10^{-5}$ | $9.91 \times 10^{-3} \pm 2.9 \times 10^{-3}$ |
| | (Vanilla, POD, PoU) | $1.91 \pm 2.4 \times 10^{-2}$ | $2.03 \times 10^{-2} \pm 4.6 \times 10^{-5}$ | $2.06 \times 10^{-2} \pm 2.3 \times 10^{-5}$ |
| | HiRefPOU-1L (POD) | $1.50 \pm 1.4 \times 10^{-2}$ | $\underline{1.56 \times 10^{-3} \pm 1.4 \times 10^{-5}}$ | $\underline{4.13 \times 10^{-4} \pm 2.7 \times 10^{-5}}$ |
| | HiRefPOU-2L (POD) | $1.69 \pm 1.4 \times 10^{-2}$ | $4.57 \times 10^{-3} \pm 1.3 \times 10^{-6}$ | $\mathbf{3.38 \times 10^{-4} \pm 1.3 \times 10^{-6}}$ |
| | HiRefPOU-3L (POD) | $1.80 \pm 4.7 \times 10^{-4}$ | $1.07 \times 10^{-2} \pm 1.1 \times 10^{-5}$ | $6.84 \times 10^{-4} \pm 3.9 \times 10^{-6}$ |
| **FNOs** | GeoFNO | $9.09 \times 10^{-1} \pm 1.5 \times 10^{-2}$ | $8.65 \times 10^{-1} \pm 1.3 \times 10^{-2}$ | $1.51 \times 10^{-1} \pm 3.4 \times 10^{-4}$ |
| | LocalFNO | $8.37 \times 10^{-1} \pm 1.3 \times 10^{-2}$ | $3.08 \times 10^{-1} \pm 1.1 \times 10^{-2}$ | $2.21 \times 10^{-2} \pm 3.8 \times 10^{-4}$ |
| | PoU-SharedFNO (LocalFNO) | $\mathbf{8.23 \times 10^{-1} \pm 1.1 \times 10^{-2}}$ | $3.22 \times 10^{-1} \pm 7.5 \times 10^{-3}$ | $2.24 \times 10^{-2} \pm 4.5 \times 10^{-4}$ |
| | PoU-ExpertFNO (LocalFNO) | $\underline{8.36 \times 10^{-1} \pm 2.4 \times 10^{-2}}$ | $4.43 \times 10^{-1} \pm 1.6 \times 10^{-2}$ | $2.85 \times 10^{-2} \pm 1.5 \times 10^{-3}$ |

and the periodic bump envelope is

$$B(\xi; c, w) = \exp\left(-\frac{\sin^2(\pi(\xi - c))}{w^2}\right). \tag{56}$$

The three packet centers are sampled hierarchically, with $c_2$ chosen near $c_1$ and $c_3$ chosen near $c_2$, so that the medium-scale packet is spatially nested inside the broad-scale packet and the fine-scale packet is nested inside the medium-scale packet. The widths and frequencies are sampled so that

$$w_1 > w_2 > w_3, \qquad k_1 < k_2 < k_3, \tag{57}$$

thereby yielding broad, medium, and fine localized oscillatory features, respectively. The amplitudes, phases, and relative signs are randomized across samples.

This benchmark is intended to test whether deeper adaptive partitioning provides measurable gains when the target operator exhibits genuinely nested structure across both space and time. A shallow partition hierarchy may localize broad active regions of the solution field, while additional refinement levels can specialize to progressively finer spatiotemporal corrections within those regions.

*Results.* Table 2 reports the testing relative $\ell_2$ errors, while Table 8 reports full-field $\ell_\infty$ errors, training times, and inference times for the KS benchmark. Within the DeepONet family, the HiRefPOU variants substantially improve over the vanilla and static-partition baselines. In particular, HiRefPOU-1L achieves the lowest DeepONet-family relative $\ell_2$ and $\ell_\infty$ errors, indicating that adaptive localization is beneficial for this dynamically unstable space–time prediction task. However, unlike the reaction–diffusion benchmarks, the strongest overall results are obtained by the FNO-based models. PoU-SharedFNO with a LocalFNO backbone achieves the lowest relative $\ell_2$ error in Table 2 and the lowest full-field $\ell_\infty$ error in Table 8, while LocalFNO and PoU-ExpertFNO are also competitive.

## 7.8 Discussion

The results reveal a problem-dependent relationship between adaptive spatial partitioning, hierarchical refinement, and the choice of operator-learning backbone. Across the DeepONet-family models, the proposed HiRefPOU architectures consistently improve over the corresponding vanilla and static-partition DeepONet baselines. This trend is especially clear on the cavity flow and reaction–diffusion benchmarks, where the learned partitions provide a mechanism for concentrating model capacity near localized flow features, coefficient interfaces, or regions of strong spatial heterogeneity.

The comparison with FNO-based models is more nuanced. On the one-dimensional periodic Burgers and Kuramoto–Sivashinsky benchmarks, LocalFNO and PoU-FNO variants are highly competitive and sometimes

achieve the lowest errors overall. This is not unexpected: these benchmarks are defined on regular periodic grids, and their solution fields contain coherent space–time structures that are well matched to Fourier-based convolutional representations. In such settings, LocalFNO provides a strong inductive bias by combining local spatial processing with spectral information, and the advantage of adaptive DeepONet partitioning is reduced. At the same time, the PoU-FNO results show that the proposed partition-of-unity mechanism can also improve Fourier-based backbones in some regimes, most notably on the Full-Field Burgers and KS benchmarks.

For the PoU-FNO variants, the backbone is selected to match the geometry and data representation of each benchmark. For the one-dimensional periodic Burgers and KS problems, we use LocalFNO as the backbone because these problems are naturally represented on regular space–time grids and local Fourier structure is a strong modeling prior. For cavity flow and the reaction–diffusion problems, we use GeoFNO as the backbone because these datasets involve geometry-dependent or point-cloud-based spatial discretizations for which GeoFNO is the more appropriate Fourier baseline. Thus, the parenthetical labels in the PoU-SharedFNO and PoU-ExpertFNO rows identify the underlying FNO architecture being augmented by the PoU mechanism. This design choice allows the PoU extension to be evaluated on top of a suitable FNO backbone rather than against a mismatched baseline.

The reaction–diffusion results also illustrate the setting in which the HiRefPOU-DeepONet architecture is most effective. In the 2D and 3D reaction–diffusion problems, the solution operators are shaped by discontinuous reaction coefficients, spatially varying diffusion, and localized gradients. Here, the hierarchical DeepONet variants achieve the best relative $\ell_2$ errors, while the FNO-based models are substantially less accurate. This suggests that adaptive spatial partitioning is particularly beneficial when the solution operator contains localized heterogeneity that is not naturally aligned with a global spectral representation.

Finally, the hierarchy does not uniformly improve performance simply by adding levels. On some benchmarks, the single-level or two-level variant is sufficient, while the three-level model gives the best results on others. This behavior is consistent with the ablation studies: the benefit of additional hierarchy depends on whether the underlying problem contains nested or localized multiscale structure that can be exploited by finer partitions. Thus, the main conclusion is not that deeper hierarchies are always better, but that hierarchical PoU refinement provides a flexible mechanism for allocating local capacity when the structure of the PDE solution operator demands it.

## 8 Conclusion

This work introduced HiRefPOU, a hierarchical partition-of-unity (PoU) framework for learning PDE solution operators with localized and multiscale structure. The method combines adaptive PoU gating with residual multi-level refinement, yielding a structured mixture-of-experts architecture in which coarse global representations are progressively corrected by locally specialized experts. Unlike purely global operator models, the proposed hierarchy provides spatially localized expressivity while preserving a smooth global prediction through the differentiable PoU formulation.

Across a diverse set of benchmark problems, HiRefPOU consistently improves over the corresponding vanilla and static-partition DeepONet baselines. The gains are most pronounced on problems with localized heterogeneity, sharp coefficient variation, or spatially concentrated error structures, such as cavity flow and the reaction–diffusion benchmarks. The learned partitions adapt to problem-specific features and provide an interpretable mechanism for allocating representational capacity to regions where the solution operator is more difficult to approximate.

The comparison with FNO-based models shows that the best-performing backbone depends on the structure of the problem. For periodic, grid-aligned space–time benchmarks such as Burgers and Kuramoto–Sivashinsky, LocalFNO and PoU-FNO variants are highly competitive and sometimes achieve the lowest errors overall. For heterogeneous reaction–diffusion problems, the HiRefPOU-DeepONet variants achieve the strongest accuracy, suggesting that adaptive spatial partitioning is especially valuable when the operator contains localized features that are not naturally captured by global spectral representations. These results indicate that the

proposed PoU mechanism is not limited to a single neural-operator family: it can improve DeepONet-style models directly and can also be used as a localization module for Fourier-based backbones.

Overall, the results support a nuanced conclusion. Hierarchical PoU refinement is not uniformly superior to every Fourier-based architecture on every benchmark, nor does increasing the number of levels always improve accuracy. Rather, the proposed framework provides a flexible and interpretable way to introduce adaptive locality into neural operator learning. When the PDE solution operator contains localized, heterogeneous, or multiscale structure, this localization can substantially improve accuracy without relying solely on increased parameter count. These findings suggest that structured, geometry-aware localization is an important ingredient for building reliable operator-learning models for complex scientific computing problems.

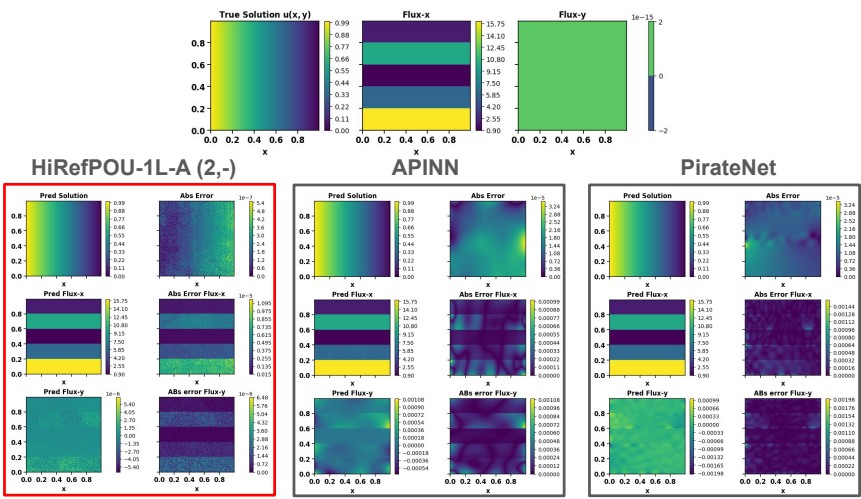

Figure 14: (Top) the true solution and fluxes. (Bottom) Predicted solutions, fluxes, and absolute errors for the Darcy flow benchmark using HiRefPOU-1L, APINN, and PirateNet. Left bottom figures show the predictions while the right bottom figures show the absolute errors.

# A   Additional PINN Experiments: Darcy Flow with Discontinuous Coefficients

This appendix provides additional technical details for the Darcy flow experiment presented in Section 7.1. These results are included for completeness and are not intended as an exhaustive comparison with domain-decomposition PINN methods such as xPINNs.

**Problem formulation and interface conditions**   We consider the steady Darcy problem

$$-\nabla\cdot\big(\mu(x)\,\nabla\phi(x)\big) = f(x), \qquad x \in \Omega = [0,1]^2, \tag{58}$$

with permeability field $\mu(x)$ that is piecewise constant across subdomains. The flux is defined as $\mathbf{q} = -\mu\nabla\phi$.

At material interfaces $\Gamma_{ij}$ between subdomains $\Omega_i$ and $\Omega_j$, the governing equations impose the physical continuity conditions

$$\phi_i(x) = \phi_j(x), \tag{59}$$
$$\mathbf{n}\cdot\mu_i\nabla\phi_i(x) = \mathbf{n}\cdot\mu_j\nabla\phi_j(x), \qquad x \in \Gamma_{ij}, \tag{60}$$

while tangential flux components may be discontinuous.

**Five-strip manufactured solution**   The domain is partitioned into five horizontal layers

$$\Omega_i = \{(x,y) \in [0,1]^2 : 0.2(i-1) \leq y \leq 0.2i\}, \quad i = 1, \dots, 5,$$

with permeability values

$$\mu|_{\Omega_1} = 16, \ \mu|_{\Omega_2} = 6, \ \mu|_{\Omega_3} = 1, \ \mu|_{\Omega_4} = 10, \ \mu|_{\Omega_5} = 2.$$

The manufactured solution $\phi_{\mathrm{ex}}(x, y) = 1 - x$ yields exact satisfaction of the interface conditions.

**Limitations of globally smooth PINNs**  A central difficulty in applying standard PINNs to interface problems arises from the use of automatic differentiation over globally smooth neural networks. When $\mu(x)$ is discontinuous, the product $\mu(x)\nabla\phi_\theta(x)$ is not differentiable across interfaces. As a result, the PDE residual computed via automatic differentiation is inconsistent near interfaces, leading to spurious residual terms and degraded accuracy (Yao et al., 2023). Because the neural approximation $\phi_\theta(x)$ is globally smooth, it cannot represent solutions whose gradients are discontinuous, which explains the oscillatory behavior and reduced flux accuracy observed for standard PINN formulations.

**Furthest Point Sampling for Center Point Initialization.**  We initialize the centers and associated radii using the farthest point sampling (FPS) and greedy covering strategy, a technique often used in meshfree approximation and RBF-based partitioning. Given a set of data points $X = \{x_1, \ldots, x_N\}$ and a desired number of centers $N_c$, the algorithm proceeds as follows. Starting from a randomly chosen initial point (usefully one on the domain boundary) $c_1 \in X$, each subsequent center $c_i$ is selected as the data point in $X$ that maximizes its minimum distance to all previously chosen centers $\{c_1, \ldots, c_{i-1}\}$. This greedy criterion ensures that each new center lies in the least represented region of the domain, resulting in a near-uniform covering with respect to the underlying data geometry.

An appealing property of the FPS initialization is that it naturally induces a coarse-to-fine hierarchical ordering among centers. Early-selected centers, which are chosen to maximize coverage over large uncovered regions, tend to have broader spatial influence and correspond to coarse partitions of the domain. Later-selected centers, in contrast, populate smaller residual gaps between existing regions, resulting in finer and more localized coverage. This ordering provides an implicit hierarchy that aligns well with our residual POU architecture: coarse centers can be associated with early network layers capturing global structure, while subsequent centers refine these representations within localized subdomains. Thus, the initialization not only ensures uniform domain coverage but also embeds a multi-resolution decomposition directly into the model's parameterization, offering a natural bridge between geometric sampling and hierarchical neural refinement. Finally, this initialization strategy is domain-agnostic and dimension-independent, since it only requires access to the input point cloud $X$.

## B  Additional Burgers Results

This section reports additional ablations for the standard 1D Burgers benchmark, focusing on the gate parameterization and the sensitivity of the 1L HiRefPOU-DeepONet model to the number of adaptive partitions.

Table 9: *Standard 1D Burgers benchmark: gate ablation for the HiRefPOU-1L DeepONet with $M = 3$ adaptive partitions. We compare RBF, box, and hybrid gate formulations. Reported values are mean $\pm$ standard deviation over random seeds. Epoch time denotes the average wall-clock time per training epoch.*

| Gate | Test rel. $\ell_2$ | Std. | Test rel. $\ell_\infty$ | Std. | Epoch time |
|------|------|------|------|------|------|
| RBF gate | 0.177134 | 0.004654 | 0.457744 | 0.009059 | 0.000542 |
| Box gate | 0.176758 | 0.004868 | 0.457380 | 0.007711 | 0.000606 |
| Hybrid gate | **0.172893** | 0.001394 | **0.454045** | 0.005357 | 0.000629 |

**Gate ablation.**  Table 9 compares three gate parameterizations for the HiRefPOU-1L DeepONet on the standard 1D Burgers benchmark with $M = 3$ adaptive partitions. Among the three variants, the hybrid gate

achieves the lowest mean test relative $\ell_2$ and $\ell_\infty$ errors, while also exhibiting the smallest standard deviation across seeds. The box and RBF gates perform similarly to one another, but both are slightly less accurate and less stable than the hybrid formulation. Although the hybrid gate incurs a modest increase in per-epoch cost, the overall trend suggests that combining center-based and box-based structure yields a more robust gate than either component alone on this benchmark.

Table 10: *Standard 1D Burgers benchmark: sensitivity of the HiRefPOU-1L DeepONet to the number of adaptive partitions M when using the hybrid gate. Reported values are mean ± standard deviation over random seeds. Epoch time denotes the average wall-clock time per training epoch.*

| Configuration | Test rel. $\ell_2$ | Std. | Test rel. $\ell_\infty$ | Std. | Epoch time |
|---|---|---|---|---|---|
| Hybrid gate, $M = 2$ | **0.172622** | 0.002183 | **0.452593** | 0.002469 | 0.000547 |
| Hybrid gate, $M = 3$ | 0.176576 | 0.001287 | 0.459151 | 0.006000 | 0.000629 |
| Hybrid gate, $M = 4$ | 0.174892 | 0.002286 | 0.458236 | 0.001768 | 0.000692 |
| Hybrid gate, $M = 6$ | 0.173398 | 0.002347 | 0.453612 | 0.006004 | 0.000833 |

**Partition-count sensitivity.**  Table 10 studies the sensitivity of the HiRefPOU-1L DeepONet to the number of adaptive partitions when using the hybrid gate. The results do not show a monotone improvement as $M$ increases. Instead, relatively small partition counts already perform well, with $M = 2$ giving the best mean test relative $\ell_2$ and $\ell_\infty$ errors in this study, and $M = 6$ achieving comparable accuracy at higher computational cost. This suggests that for the standard 1D Burgers benchmark, the main benefit comes from introducing adaptive localization rather than from aggressively increasing the number of partitions.

## C  Additional Cavity Ablations

This section reports additional ablations for the cavity benchmark, including the sensitivity of the 1L model to the partition count and residual strength, the effect of gate parameterization, and the effect of child allocation in the 2L hierarchy.

Table 11: *Cavity benchmark: partition-count and residual-strength ablation for the HiRefPOU-1L model. The parameter $\alpha$ denotes the residual scaling used in the POU correction term. Reported values are mean ± standard deviation over random seeds.*

| Gate | $M$ | $\alpha$ | Final test rel. $\ell_2$ | Std. |
|---|---|---|---|---|
| hybrid | 6 | 1 | $1.298 \times 10^{-3}$ | $1.858 \times 10^{-4}$ |
| hybrid | 6 | 1.25 | $1.385 \times 10^{-3}$ | $8.321 \times 10^{-5}$ |
| hybrid | 6 | 0.75 | $1.418 \times 10^{-3}$ | $1.589 \times 10^{-4}$ |
| hybrid | 2 | 1.25 | $1.996 \times 10^{-3}$ | $3.662 \times 10^{-4}$ |
| hybrid | 2 | 1 | $1.868 \times 10^{-3}$ | $7.134 \times 10^{-5}$ |
| hybrid | 2 | 0.75 | $1.984 \times 10^{-3}$ | $1.467 \times 10^{-4}$ |

**Partition-count and residual-strength ablation.**  Table 11 examines the effect of the partition count $M$ and the residual scaling parameter $\alpha$ for the HiRefPOU-1L model on the cavity benchmark. Across both values of $M$, the choice $\alpha = 1$ gives the best mean test relative $\ell_2$ error, while either increasing or decreasing the residual strength leads to a mild degradation in performance. The impact of the partition count is more pronounced: increasing from $M = 2$ to $M = 6$ substantially improves accuracy for all three $\alpha$ values. These

results suggest that the cavity problem benefits more strongly from increased spatial specialization than the Burgers benchmark, while also indicating that the unscaled residual contribution ($\alpha = 1$) provides the most reliable 1L setting among the values tested here.

Table 12: *Cavity benchmark: gate-type ablation for the HiRefPOU-1L model. Reported values are mean $\pm$ standard deviation over random seeds.*

| Gate | Final test rel. $\ell_2$ | Std. |
|------|--------------------------|------|
| box | $1.559 \times 10^{-3}$ | $3.561 \times 10^{-4}$ |
| hybrid | $1.536 \times 10^{-3}$ | $1.588 \times 10^{-4}$ |
| rbf | $1.821 \times 10^{-3}$ | $2.109 \times 10^{-4}$ |

**Gate-type ablation.** Table 12 compares the box, hybrid, and RBF gate parameterizations for the HiRefPOU-1L model on the cavity benchmark. The hybrid gate achieves the best mean test relative $\ell_2$ error, though its advantage over the box gate is modest. In contrast, the RBF gate performs noticeably worse than both the box and hybrid variants. Taken together with the Burgers results, this supports the use of the hybrid gate as a consistent default choice: it is never worse in these experiments and is generally either the best or tied for best, while also offering lower variability than the purely RBF formulation.

Table 13: *Cavity benchmark: child-allocation ablation for the HiRefPOU-2L model. Here, $M_c$ and $M_f$ denote the numbers of coarse and fine experts, respectively, while $\alpha_c$ and $\alpha_f$ denote the corresponding residual strengths. The allocation label indicates how fine-level children are distributed across the coarse-level parents. Reported values are mean $\pm$ standard deviation over random seeds.*

| Allocation | $M_c$ | $M_f$ | $\alpha_c$ | $\alpha_f$ | Final test rel. $\ell_2$ | Std. |
|------------|-------|-------|------------|------------|--------------------------|------|
| 0_2 | 2 | 2 | 1 | 1.25 | $1.955 \times 10^{-3}$ | $1.827 \times 10^{-4}$ |
| 0_2 | 2 | 2 | 1 | 0.75 | $1.920 \times 10^{-3}$ | $1.812 \times 10^{-4}$ |
| 2_0 | 2 | 2 | 1 | 1 | $2.013 \times 10^{-3}$ | $3.089 \times 10^{-4}$ |
| 2_2 | 2 | 4 | 1 | 1 | $1.944 \times 10^{-3}$ | $1.421 \times 10^{-4}$ |
| 2_0 | 2 | 2 | 1 | 0.75 | $2.012 \times 10^{-3}$ | $1.736 \times 10^{-4}$ |
| 2_0 | 2 | 2 | 1 | 1.25 | $2.123 \times 10^{-3}$ | $2.716 \times 10^{-4}$ |
| 0_2 | 2 | 2 | 1 | 1 | $1.934 \times 10^{-3}$ | $1.780 \times 10^{-4}$ |
| 2_2 | 2 | 4 | 1 | 1.25 | $1.966 \times 10^{-3}$ | $1.728 \times 10^{-4}$ |
| 2_2 | 2 | 4 | 1 | 0.75 | $1.971 \times 10^{-3}$ | $1.695 \times 10^{-4}$ |

**Child-allocation ablation.** Table 13 studies how the distribution of fine-level children across the coarse parents affects performance in the HiRefPOU-2L cavity model. The results show that performance is sensitive to how refinement capacity is allocated. Among the tested configurations, the `0_2` allocation yields the best results, while the `2_0` allocation is consistently the weakest. The balanced `2_2` configuration performs competitively, but does not outperform the best asymmetric allocation in this experiment. This indicates that, for the cavity benchmark, refinement is most effective when it is concentrated in the parent region associated with the more challenging local structure rather than distributed uniformly by default.

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
