# OpenReview forum: "Localized Operator Learning with Adaptive Partition-of-Unity Mixture-of-Expert Networks"
_TMLR — Under review for TMLR_

### Review · Reviewer_L1CG · 2026-06-25

**Summary Of Contributions:**

The paper proposes a localized neural-operator framework based on adaptive partition-of-unity mixture-of-experts models. The main contribution is HiRefPOU, a hierarchical residual POU architecture for DeepONets, where the coordinate-dependent trunk representation is decomposed into a global trunk plus coarse and fine local expert corrections. The local experts are blended using smooth POU gates, so the model preserves global continuity while allowing different spatial regions to use specialized trunk bases. The paper also introduces a geometry-aware gating mechanism combining center-distance/RBF-like gating with box-initialized residual gating, and it explores one-, two-, and three-level hierarchical variants. The authors further show that a similar POU mechanism can be incorporated into FNO-type architectures through POU-SharedFNO and POU-ExpertFNO variants.

I should note that my own background is closer to the INR community than to the PINN community. From this perspective, I understand the proposed method as taking a fairly natural localized-coordinate-representation idea: global coordinate network plus local experts, smooth masks, and coarse-to-fine residual refinement. In that sense, the core idea is intuitive rather than surprising. However, the paper makes this idea concrete in a neural-operator setting and provides a systematic empirical study showing when this inductive bias is useful.

A key strength of the work is that the proposed inductive bias is well matched to PDE operator-learning problems with localized heterogeneity, coefficient discontinuities, sharp interfaces, or spatially concentrated error structures. The empirical results show consistent improvements over vanilla DeepONet and static POU DeepONet baselines.

The main weakness is that the conceptual novelty appears somewhat incremental when viewed from the INR perspective: local experts, smooth blending masks, and residual coarse-to-fine refinement are fairly standard design principles. Moreover, the current implementation evaluates all experts densely at all query coordinates, so it does not obtain computational savings from sparse routing and can be several times more expensive than simpler DeepONet baselines. The gains are also problem- and backbone-dependent. Finally, the gating depends only on spatial coordinates rather than the input function, which may limit applicability when localized difficult regions move from sample to sample.

**Audience:**

Yes

**Audience Explanation:**

I expect the paper to be of interest to a subset of the TMLR audience working on neural operators and coordinate-based neural representations. Even though I personally approach the paper from an INR perspective rather than a PINN/PDE-solver perspective, I found the main idea easy to interpret as a localized coordinate-representation mechanism applied to the DeepONet trunk. This connection may also be useful to readers interested in the relationship between implicit neural representations and neural operator architectures.

**Broader Impact Concerns:**

N/A in this case.

**Claims And Evidence:**

Yes

**Claims Explanation:**

The main claims are supported by a reasonably comprehensive experimental study. The paper claims that adaptive POU localization can improve DeepONet-style operator learning for PDE families with localized heterogeneity or sharp structure, and this is well supported by the cavity flow and reaction–diffusion results. In these settings, HiRefPOU substantially reduces relative error compared with vanilla DeepONet, overparameterized global DeepONet, and static POU/POD-POU DeepONet baselines. The visualizations of learned partitions further support the interpretability claim, since the gates tend to align with meaningful spatial regions such as coefficient interfaces or areas of strong solution variation.

**Requested Changes:**

Critical:
- The paper would benefit from explicitly positioning the contribution as a localized neural operator architecture rather than implying a fundamentally new modeling paradigm. From the perspective of coordinate-based neural representations, the overall idea of combining global and local coordinate representations with smooth spatial gating is intuitive, and I believe the paper would be stronger if this positioning were made more explicit.
- Although the paper occasionally describes the model as routing computation to specialized experts, the current implementation evaluates all experts densely and combines them through POU weights. I recommend emphasizing this distinction more clearly to avoid readers overinterpreting the computational advantages of the approach.

Minor:
- Since the gating depends only on spatial coordinates rather than the input function, it would be valuable to evaluate problems where discontinuities or regions of interest are input-dependent. This would better characterize the limitations of coordinate-only gating.
- Several of the strongest results use POD augmentation. Additional ablations separating the effects of adaptive POU hierarchy and POD would make the empirical conclusions clearer.
- The experiments suggest that different numbers of hierarchy levels and experts are preferable for different PDE families. A short discussion or empirical guideline on choosing these hyperparameters would improve the paper's practical value.
- Since the current implementation already provides adaptive localization, it would be interesting to comment on whether sparse expert activation or approximate routing could be incorporated in future work to improve computational efficiency.

---

### Review · Reviewer_uaii · 2026-07-02

**Summary Of Contributions:**

This paper proposes HiRefPOU, a localized neural-operator architecture that combines partition-of-unity (PoU) mixture-of-experts gating with residual coarse-to-fine refinement. The main instantiation is a DeepONet variant in which the trunk representation is decomposed into global, coarse local, and fine local components, with smooth PoU gates ensuring continuous blending across spatial regions. The paper also explores PoU extensions of FNO-style backbones, where learned spatial gates blend localized expert heads or expert operators.
The central empirical claim is that explicit learned spatial localization improves neural-operator accuracy and interpretability on PDE families with localized heterogeneity, sharp coefficient changes, or multiscale structure. The paper evaluates this claim on a range of benchmarks, including cavity flow, 2D and 3D reaction-diffusion, Burgers variants, Kuramoto-Sivashinsky, and a motivating Darcy-flow PINN case study. The results are appropriately nuanced: HiRefPOU is consistently stronger than vanilla and static-PoU DeepONet baselines, especially on heterogeneous reaction-diffusion problems, while FNO-based models remain stronger on some regular periodic grid benchmarks.

Key strengths include: (i) a coherent architectural idea that connects PoU approximation, mixture-of-experts localization, and residual hierarchical refinement; (ii) broad empirical coverage across several PDE/operator-learning settings; (iii) useful comparisons against vanilla DeepONets, overparameterized DeepONet baselines, static PoU variants, GeoFNO, LocalFNO, and PoU-FNO variants; (iv) timing results and ablations that expose accuracy-cost tradeoffs; and (v) qualitative partition visualizations that help explain where the learned local experts specialize.

The main weaknesses are: (i) the word "adaptive" is somewhat underspecified, since the gates appear to be learned coordinate-dependent partitions shared across the dataset rather than input-conditioned partitions that adapt per PDE instance; (ii) the evidence for hierarchy specifically is mixed, with 1L sometimes matching or outperforming 2L/3L, so hierarchy-related claims should be carefully scoped; (iii) parameter counts, hyperparameter budgets, and baseline tuning details are not always explicit enough to fully assess fairness; (iv) interpretability is mostly qualitative, without a quantitative measure of alignment between learned partitions and physical interfaces or high-error regions; and (v) reproducibility would benefit from more complete dataset, code, and configuration details.

**Audience:**

Yes

**Audience Explanation:**

The paper should be of interest to a meaningful subset of the TMLR audience working on neural operators, scientific machine learning, mixture-of-experts architectures, and interpretable/localized approximation methods. It addresses a real limitation of global DeepONet and FNO-style models: localized heterogeneity and sharp interfaces can be difficult to represent with purely global bases or spectral layers. The proposed PoU-MoE formulation is a natural and useful architectural direction for this problem.

The findings are also useful beyond the specific architecture. The paper provides evidence that the value of localization depends strongly on the PDE structure and the chosen backbone. This is a helpful empirical message for the community: local/hierarchical models are especially useful for heterogeneous coefficient/interface problems, while Fourier-local models can be preferable on regular periodic space-time benchmarks. Even if some contributions are incremental relative to existing PoU-MoE DeepONets and localized FNOs, the combination of hierarchical residual PoU refinement, geometry-aware gates, and broad benchmark comparison provides enough insight to be relevant to TMLR readers.

**Broader Impact Concerns:**

I do not see major broader-impact concerns that would block publication. The work is primarily methodological and targets surrogate modeling for PDE-governed scientific computing. However, because neural operators may eventually be used in engineering or scientific decision pipelines, the authors should briefly caution that these models are not certified numerical solvers and should not be used in safety-critical settings without validation, uncertainty quantification, and problem-specific error checks. If the submission does not already include a broader impact or limitations statement, a short paragraph covering these deployment caveats would be appropriate.

**Claims And Evidence:**

Yes

**Claims Explanation:**

Overall, the main claims are supported, provided they are read with the nuance already present in parts of the paper. The strongest supported claim is that learned PoU localization improves DeepONet-family models over vanilla, overparameterized, and static-PoU baselines on several PDE operator-learning tasks. The reaction-diffusion and cavity-flow experiments are especially convincing: the reported relative errors and infinity-norm errors favor HiRefPOU variants within the DeepONet family, and the partition visualizations show plausible spatial specialization near interfaces or localized flow structures.

The paper is also careful to show that the method is not uniformly superior to all neural-operator backbones. On periodic grid-aligned Burgers and KS benchmarks, LocalFNO or PoU-FNO variants are often stronger, and the discussion correctly attributes this to the problem structure and the Fourier/local convolutional inductive bias. This makes the empirical story more credible than a blanket "our method is best" claim.

That said, several claims require clarification or tempering. First, "adaptive partitioning" should be defined precisely. As written, the gating networks appear to learn spatial partitions during training, but the gates depend on coordinates rather than on the input function or PDE instance. This supports dataset-level geometric adaptivity, but not necessarily instance-wise adaptivity to moving interfaces or sample-dependent localized structures. Second, the evidence for deeper hierarchy is problem-dependent: 2L/3L improve on some reaction-diffusion settings, but 1L is best or essentially tied on others. Claims about hierarchy should therefore emphasize conditional benefit rather than general superiority. Third, the paper reports means and standard deviations, but some improvements are modest relative to variability; statistical significance or more careful wording would strengthen the evidence. Finally, interpretability is supported by useful qualitative plots, but the claim that partitions align with physical structures would be more convincing with a quantitative alignment or error-localization analysis.

**Requested Changes:**

Critical: Clarify the meaning and scope of "adaptive" gating. The current formulation appears to use coordinate-dependent gates that are learned during training and then shared across all samples. This is a valid and useful form of learned geometric partitioning, but it is different from input-conditioned or instance-adaptive routing. The authors should explicitly state this distinction and avoid language suggesting per-instance adaptivity unless such gates are actually used. If the intended claim is broader, please add an experiment with moving interfaces, sample-dependent coefficient discontinuities, or input-conditioned localized structures.

Critical: Tighten claims about hierarchy. The experiments show that additional hierarchy helps on some benchmarks, especially reaction-diffusion settings, but 1L is competitive or best on others. The abstract, introduction, and conclusion should consistently say that hierarchy is beneficial when the operator contains exploitable nested/localized multiscale structure, not that deeper refinement is generally superior. Where differences are within standard deviation, please avoid strong comparative claims.

Critical: Provide clearer evidence that gains are not mainly due to parameter count, implementation efficiency, or tuning budget. The paper includes a useful (P+1)-Vanilla DeepONet baseline, but the reader needs explicit parameter counts, training budgets, hyperparameter search ranges, and model-size comparisons for all major baselines and HiRefPOU variants. This is especially important because all experts are evaluated densely, so the architecture increases computation rather than providing sparse routing savings. Relatedly, some reported training times are lower than competing baselines despite the use of multiple densely evaluated experts. The authors should clarify whether this is due to smaller hidden dimensions, vectorized GPU execution, differences in parameter counts, the computational cost of GeoFNO/LocalFNO baselines, or other implementation details. Reporting per-iteration time, FLOPs, memory use, and error-vs-wall-clock-time curves would make the efficiency comparison much more interpretable.

Critical: Strengthen reproducibility details. Please include complete architecture tables, partition counts, expert widths/depths, POD dimensions, optimizer schedules, random seeds, dataset generation procedures, train/test splits, solver settings, and baseline hyperparameters. If code is provided, it should include scripts/configs sufficient to reproduce each table, ideally in an anonymized form appropriate for the review process.

Critical: Better separate novelty from closely related work. The paper should more explicitly distinguish HiRefPOU from existing PoU-MoE DeepONets, partition-of-unity networks, hierarchical PoU approximation, APINNs/XPINNs, HyResPINNs, GeoFNO, and LocalFNO. A concise table comparing fixed vs learned gates, flat vs hierarchical structure, operator-learning vs PINN settings, input-conditioned vs coordinate-only routing, and sparse vs dense expert evaluation would help.

Important but not necessarily acceptance-critical: Quantify interpretability claims. The learned partitions are visually plausible, but the paper should include at least one quantitative measure, such as gate/interface alignment, partition entropy, distance from high-gradient or high-error regions, or the fraction of error reduction concentrated near physical discontinuities.

Important but not necessarily acceptance-critical: Clarify computational cost and routing language. Since the current implementation evaluates all experts densely, phrases such as "route computation to specialized local experts" may be read as implying sparse MoE-style computational routing. Please state clearly that the present experiments use dense expert evaluation, and discuss what would be required for sparse routed inference. The training-time tables should also be interpreted carefully: lower wall-clock time for some HiRefPOU variants does not by itself establish algorithmic efficiency unless the model sizes, implementation details, and per-step costs are normalized across baselines.

---

### Review · Reviewer_wgmp · 2026-07-22

**Summary Of Contributions:**

This paper proposes HiRefPOU, a localized neural-operator framework that combines adaptive partition-of-unity (POU) gating, mixture-of-experts, and additive hierarchical refinement. The main architectural contribution is a DeepONet variant in which the global trunk representation is augmented by coarse and fine spatial expert mixtures, with child partition weights constrained by their parent partitions to preserve a smooth global POU representation. The paper also introduces a geometry-aware hybrid gate that combines learnable center-based distance features with a deformable neural gate, and demonstrates that the same POU principle can be incorporated into FNO-type models without changing their spectral layers. The method is evaluated on cavity flow, Burgers, Kuramoto–Sivashinsky, and two- and three-dimensional reaction–diffusion problems, together with a motivating Darcy-flow case study. The results show that adaptive POU localization consistently improves over global and fixed-partition DeepONet baselines, with the largest gains on problems containing fixed interfaces, heterogeneous coefficients, or localized gradients. Comparisons with LocalFNO also indicate that the benefit is problem- and backbone-dependent: Fourier-based local models remain stronger on several regular periodic-grid tasks.

Key strengths include the clear motivation, modular architecture, broad empirical evaluation, inclusion of parameter-enlarged and locality-aware baselines, and interpretable spatial partitions that often align with difficult physical regions. A main limitation is that the hierarchy is manually specified and primarily guarantees nested spatial routing, rather than a strict coarse-to-fine decomposition of the learned functions. In particular, the formulation does not explicitly enforce frequency separation, orthogonality, or stagewise residual specialization between coarse and fine experts, and deeper hierarchies do not consistently improve performance. The current dense evaluation of all experts also introduces additional computational cost and does not exploit sparse routing.

**Audience:**

Yes

**Audience Explanation:**

At least some members of the TMLR audience would be interested in these findings because the paper addresses an active intersection of neural operator learning, mixture-of-experts architectures, and scientific machine learning. Beyond proposing a new localized DeepONet design, the empirical study provides useful evidence about when explicit spatial localization is beneficial: adaptive POU models perform particularly well for PDE operators with fixed interfaces, heterogeneous coefficients, and localized gradients, whereas locality-aware Fourier models remain stronger on several regular, periodic-grid problems. This problem-dependent comparison is valuable even to readers who may not adopt the proposed architecture directly, since it clarifies how geometric localization interacts with different operator-learning backbones and highlights practical accuracy–cost tradeoffs. The learned partitions and their alignment with physically difficult regions may also interest researchers working on interpretable expert routing, adaptive numerical approximation, and PDE surrogate modeling.

**Broader Impact Concerns:**

I did not identify any broader-impact concerns specific to this work that would require an additional Broader Impact Statement. The paper presents a methodological contribution to neural-operator approximation for PDEs and evaluates it on standard scientific-computing benchmarks; it does not involve human participants, personal or sensitive data, socially consequential decision-making, or other immediately concerning applications.

**Claims And Evidence:**

Yes

**Claims Explanation:**

The claims made in the submission are generally supported by accurate, convincing, and clearly presented evidence. The evaluation covers a diverse set of PDE operator-learning problems, includes global, overparameterized, fixed-POU, and locality-aware FNO baselines, and reports relative $\ell\_2$, $\ell\_\inf$, computational cost, and variability across five random seeds. In particular, the results consistently support the central claim that adaptive POU localization improves DeepONet-based models over their global and static-partition counterparts, with especially strong gains for problems involving coefficient discontinuities, fixed interfaces, and localized spatial heterogeneity. The error-field and gating visualizations further provide intuitive evidence that the learned partitions often align with physically difficult regions. The manuscript is also appropriately transparent about problem dependence: deeper hierarchies are not uniformly superior, and LocalFNO-based models perform better on several periodic, grid-aligned benchmarks. However, some stronger mechanistic statements about a strict coarse-to-fine or multiresolution decomposition remain only partially justified. The nested POU construction guarantees parent–child spatial routing and smooth aggregation, but it does not explicitly enforce frequency separation, orthogonality, or stagewise residual specialization between coarse and fine experts. Level-wise contribution or spectral analyses would therefore be useful to substantiate these particular claims more fully.

**Requested Changes:**

Overall, I view the paper as close to publishable. I consider the first item below important for securing my recommendation for acceptance, while the remaining items would strengthen the interpretation and presentation of the work rather than being strict conditions for acceptance.

1. Critical for acceptance — Clarify the scope of the hierarchical residual and coarse-to-fine claims.

The manuscript repeatedly describes the proposed architecture as a residual coarse-to-fine or multiresolution decomposition in which each level focuses on structure not resolved by the preceding levels. However, the nested POU formulation directly guarantees parent–child spatial routing, smooth aggregation, and global continuity, but does not appear to explicitly enforce frequency separation, orthogonality, or stage-wise residual specialization between the coarse and fine experts. I therefore recommend either moderating statements that imply a strict multiresolution decomposition or providing further justification for them. The authors should also state the complete training objective more explicitly, particularly the “hierarchical residual losses” mentioned in Section 6.3, and clarify whether any staged training, freezing, or stop-gradient mechanism is used.

2. Would strengthen the work — Provide more direct visual or quantitative evidence of hierarchical specialization.

The current partition visualization in Figure 9 does not make the distinction between coarse and fine representations especially clear: the fine-level gates appear largely as subdivisions of the coarse regions, rather than exhibiting an immediately recognizable increase in localization or resolution. This is understandable from the construction $\phi_{j,k}^f = \phi_j^c\psi_{j,k}$, but it makes it difficult to assess whether the two levels learn functionally distinct corrections. The figure could be improved by using comparable color scales, overlaying the known coefficient interface, and displaying the weighted expert contributions or the separate global, coarse, and fine output components for the same test example. A simple level-removal, correlation, spatial-gradient, or spectral analysis would also help establish whether the fine level predominantly corrects localized errors left by the coarse level.

3. Would strengthen the work — Make the hierarchy-specific comparison easier to assess.

The comparisons among the 1L, 2L, and 3L models are useful, particularly where the models use comparable partition budgets. A compact table reporting the number of experts at each level, the number of leaf partitions, trainable parameter counts, approximate computational cost, and inference time would make these comparisons more transparent. Where feasible, comparison with a flat adaptive POU model matched in leaf count and computational budget would further isolate the benefit of organizing the experts hierarchically rather than simply increasing or redistributing model capacity.

4. Would strengthen the work — Expand the discussion of the interaction with locality-aware FNO backbones.

The LocalFNO and POU-FNO results are informative and show that explicit POU localization is complementary to a local Fourier backbone on some benchmarks, but potentially redundant on others. For example, LocalFNO remains stronger on the standard periodic Burgers problem, whereas adding POU improves the LocalFNO-based model on the full-field Burgers and KS benchmarks. A clearer discussion of when geometric POU routing is expected to add value beyond an already local or multiscale backbone would help readers understand the practical scope of the proposed method. This could primarily be addressed through interpretation of the existing results and would not necessarily require substantial additional experiments.